

# Vertical mobility of pyrogenic organic matter in soils: A column experiment

Marcus Schiedung[1], Severin-Luca Bellè[1], Gabriel Sigmund[2], Karsten Kalbitz[3] and Samuel Abiven[1]

[1]Department of Geography, University of Zurich, Winterthurerstrasse 190, 8057 Zurich, Switzerland
[2]Department of Environmental Geosciences, University of Vienna, Althanstrasse 14 UZA II, 1090 Vienna, Austria
[3]Institute of Soil Science and Site Ecology, Technische Universität Dresden, Pienner Straße 19, 01737 Tharandt, Germany

*Correspondence to*: Samuel Abiven (samuel.abiven@geo.uzh.ch)

**Abstract** Pyrogenic organic matter (PyOM) is a major and persistent component of soil organic matter but its mobility and cycling in soils is largely unknown. We conducted a column experiment with a topsoil and subsoil of a sand and a sandy loam
to study the mobility of highly $^{13}$C labelled ryegrass PyOM (>2.8 at%), applied as a layer on a 7 cm long soil column, under saturated conditions. Further, we used fresh and oxidized PyOM (accelerated aging with $H_2O_2$) to identify changes in its migration through the soil with aging and associated surface oxidation. Due to the isotopic signature, we were able to trace the PyOM carbon (PyOM-C) in the soil columns, including density fractions, its effect on native soil organic carbon (nSOC) and its total export in percolates sequentially sampled after 1,000-18,000 l m$^{-2}$. In total, 4-11 % of the added PyOM-C was
mobilized and <1 % leached from the columns. The majority of PyOM-C was mobilized with the first flush of 1,000 l m$^{-2}$ (51-84 % of exported PyOM-C), but its export was on-going for the sandy soil and the loamy subsoil. Oxidized PyOM showed a 2-7 times higher mobility than fresh PyOM. In addition, twofold higher quantities of oxidized PyOM-C were leached from the sandy soil compared to the loamy soil. Besides the higher mobility of oxidized PyOM, its retention in both soils increased due to an increased reactivity of the oxidized PyOM surfaces and enhanced the interaction with the soil mineral phase. Density
fractionation of the upper 0-2.3 cm, below the PyOM application layer, revealed that up to 40 % of the migrated PyOM was associated to the mineral phase in the loamy soil, highlighting the importance of mineral interaction for the long-term fate of PyOM in soils. The nSOC export from the sandy soil significantly increased by 48-270 % with addition of PyOM compared to the control while no effect was found for the loamy soil after the whole percolation. Due to its high sorption affinity towards the soil mineral phase, PyOM can mobilize mineral-associated soil organic matter in coarse textured soils, where organo-
mineral interactions are limited, while finer textured soils have the ability to re-adsorb the mobilized soil organic matter. Our results show, that the vertical mobility of PyOM in soils is limited to a small fraction. Aging (oxidation) increases this fraction but also increases the PyOM surface reactivity and thus its long-term retention in soils. Moreover, the migration of PyOM affects the cycling of nSOC in coarse soils and thus influences the carbon cycle of fire affected soils.



## 1 Introduction

Pyrogenic organic matter (PyOM) is a product of incomplete combustion of vegetation biomass during wildfires. It is one of the oldest terrestrial organic carbon (C) pools with residence times on the order of 10,000 years and an annual global production of 196-349 Tg C (Coppola et al., 2018; Jones et al., 2019; Masiello and Druffel, 1998). After a wildfire, around 12-27 % of the initially burned biomass can remain as PyOM on the site of burning (Santín et al., 2015). It can directly enter the soils or can be transported laterally via mass transport prior to incorporation. Mass transport of PyOM mainly occurs during

the first rain event after a fire, resulting in a translocation and re-deposition within the landscape and eventually in a PyOM burial at depositional sites (Abney et al., 2019; Cotrufo et al., 2016; Rumpel et al., 2015). In addition to this natural wildfire derived PyOM, artificially produced biochar is used as agricultural soil amendment to improve soil properties and is recognized as a C sequestration strategy to mitigate climate change (Lehmann, 2007).

Between the continuum of terrestrial and aquatic ecosystems, PyOM is continuously exported from soils to rivers with

less seasonality compared to non-fire-derived soil organic matter (Dittmar et al., 2012a; Hockaday et al., 2007; Wagner et al., 2018). Globally, dissolved PyOM contributes to 0.1-15 % of the total riverine and marine dissolved organic carbon (DOC) pool (Coppola and Druffel, 2016; Jones et al., 2020). Furthermore, the quantity of dissolved PyOM in rivers was found to be decoupled from the fire history of the watershed (Ding et al., 2013; Santos et al., 2017), which indicates that soils are an important intermediate storage for PyOM in the terrestrial system prior to its export to aquatic ecosystems (Abiven and Santín,

2019; Bird et al., 2015; Santín et al., 2016).

In soils, PyOM contributes globally to 14 % (0-60 %) of the total soil organic carbon (SOC) which makes it to one of the main components of organic matter (OM) in soils (Reisser et al., 2016). Pyrogenic organic matter is found with residence times of >1000 years, which is much more than the average age of SOC and is mainly attributed to its highly condense and aromatic composition and thus increased stability against degradation (Kuzyakov et al., 2014; Schmidt et al., 2011; Singh et al., 2012).

Within the soil profile, PyOM contents can increase with depth indicating that vertical transport determines its long-term fate in soils and terrestrial ecosystems (Foereid et al., 2011; Hobley, 2019; Leifeld et al., 2007; Soucémarianadin et al., 2019).

Pyrogenic organic matter and non-fire-derived OM differ in biological (e.g. degradability) and physico-chemical interactions in soils (Bird et al., 2015; Pingree and DeLuca, 2017). Condensation reactions during pyrolysis result in a high porosity and large surface areas depending on the fuel biomass, duration and production temperature (Hammes and Abiven,

2013; Preston and Schmidt, 2006). The chemical composition, physical properties and the particle size control the mobility in soils and the interactions of PyOM with the soil mineral phase which further depend on soil properties such as texture, pH and Fe/Al-(hydr-)oxides content (Hobley, 2019; Pignatello et al., 2015). In addition, PyOM is interacting with non-fire-derived OM and influences its mobility by direct sorption and stabilization on PyOM surfaces (Jiang et al., 2019; Mukherjee and Zimmerman, 2013). However, due to its high molecular weight (rich in aromatic compounds), PyOM has a strong sorption

affinity to the soil mineral phase. This can result in a mobilization of non-fire-derived and less adsorbing OM that is already adsorbed on the soil mineral surfaces (Jiang et al., 2016; Oren and Chefetz, 2012; Zhang et al., 2017). Biotic and abiotic





oxidation alter the surface reactivity of PyOM by increasing the abundance of oxygen and hydrogen containing functional groups (e.g. carboxylic groups) and thus influencing its properties over time and with aging. Aging of PyOM is reported to enhance its water solubility and thus its mobility in soils with vertical percolation (Abiven et al., 2011; Wagner et al., 2017).

However, only a small fraction of the total PyOM (<1 %) was found to be solubilized and transported in soils (Abiven et al., 2011; Maestrini et al., 2014; Major et al., 2010). The quantities and drivers controlling the vertical PyOM mobility in dissolved or particulate form in soils, its effect on non-fire derived SOC during its migration and the influence of aging are not well understood so far. This is limiting our understanding of the fate of PyOM in the terrestrial C cycle.

In this study, we assessed the vertical PyOM mobility by conducting saturated soil column experiments (7 cm length) with

flow interruption. To trace mobilized PyOM, we applied highly [13]C labelled (>2.8 at% in excess) ryegrass PyOM-carbon (PyOM-C) on a topsoil and subsoil of a sand and sandy loam and determined the effect of soil properties on the PyOM mobility. Moreover, we compared the vertical mobility of fresh and oxidized PyOM (accelerated aging with $H_2O_2$) to identify changes in its mobility with aging and associated surface oxidation. Using highly [13]C labelled ryegrass PyOM under these controlled conditions allowed us, on the one hand, to trace even small proportions of mobilized PyOM-C in soils and potentially exported

to aquatic ecosystems, and on the other hand, to detect changes in the native soil organic carbon (nSOC) mobility. We hypothesised that, (i) PyOM is continuously exported from the soil but its rate decreases over time, (ii) a higher degree of oxidation (aging) increases the mobility of PyOM through soils, (iii) PyOM can be retained in soils during its migration and (iv) PyOM migration through the soil influences the nSOC mobility.

## 2 Materials and methods

### 2.1 Soils

Topsoils (0-10 cm) and subsoils (40-60 cm) of a sand and a sandy loam, hereafter sandy and loamy soil, were used to obtain a range of texture and SOC contents between 2.64-21.91 g C kg$^{-1}$ (Table 1). The sandy soil was sampled from an entic podzol near Gifhorn (55°22'44.0''N; 10°25'224.5''), Germany, under pine forest (*Pinus sylvestris*). The loamy soil was sampled from a haplic luvisol east of Eiken (47°32'33.5''N; 8°00'31.5''E), Switzerland, under mixed forest dominated by

beech (*Fagus sylvatica*). Both sites are similar in terms of mean annual temperature (sandy soil=8.8°C and loamy soil=10.0°C) and mean annual precipitation (sandy soil=620 mm and loamy soil=780 mm). All soil samples were dried at 40°C over night and sieved to <2 mm.

The pH was lower in the sandy topsoil (3.4 ± 0.1) compared to the loamy topsoil (5.3 ± 0.1) but similar in both subsoils with 4.1 ± 0.1 and 4.0 ± 0.1, respectively. Oxalate extractable amorphous Fe and Al (hydr)oxides (Fe(o) and Al(o))

contents were higher in the loamy soil compared to the sandy soil (Table 1). The soils further differed in the SOC distribution between free particulate organic matter (fPOM) and mineral associated organic matter (MAOM) obtained by density fractionation (see Sect. 2.4 for procedure). The MAOM fraction contained 82.8 ± 0.2 % and 85.7 ± 1.0 % of the SOC in the



loamy topsoil and subsoil, respectively. The sandy topsoil contained $73.2 \pm 0.6$ % of the SOC as fPOM. In the sandy subsoil, the fPOM contributed to $41.1 \pm 0.8$ % of the SOC.

## 2.2 $^{13}$C labelled pyrogenic organic matter

Highly $^{13}$C labelled ryegrass (*Lolium Perenne* L.) was produced in the multi-isotope controlled environment facility at the University of Zurich (Studer et al., 2017). The ryegrass was oven dried at 40°C and pyrolysed at 450°C for 4 h under $N_2$ atmosphere (Hammes et al., 2006). Three independent growing batches of the initial ryegrass were charred separately (Table 2). Artificially altered PyOM was produced by chemical and heat accelerated oxidation presented by Cross and Sohi (2013); in brief, 0.1 mol of $H_2O_2$ were added to 1 g of C and oxidized for two days at 80°C while the samples were gently shaken five to seven times a day to ensure a homogeneous oxidation. The samples were dried at 105°C over night. The oxidation was conducted in glass test tubes (20 cm length and 2.5 cm diameter) with 1 g of PyOM using a 5 % $H_2O_2$ solution. These test tubes ensured a continuous exposure of PyOM to the oxidant. The oxidized PyOM from five test tubes was homogenized and stored in a desiccator.

Mid-infrared spectra at wave lengths from 4000-400 cm$^{-1}$ were recorded (average of 64 scans per sample at 4 cm$^{-1}$ resolution) by using diffuse reflectance infrared Fourier-transformed (DRIFT) spectroscopy (TENSOR 27 spectrophotometer, Bruker, Fällanden, Switzerland). Figure 1 shows the differences in the spectra of oxidized and fresh PyOM for the three PyOM, indicating an overall increase of O- and H-containing functional groups after the PyOM oxidation (increases of absorbance at A, C, E and G; Chatterjee et al., 2012; Keiluweit et al., 2010; Wood, 1988), loss of aliphatic compounds (decrease of absorbance at B and H; Chatterjee et al., 2012; Keiluweit et al., 2010), a shift in aromaticity (increase at C and decrease at D) and decarbonation (decreases at D and H; Rechberger et al., 2017), see Fig S2 for all spectra.

## 2.3 Soil column set up and percolation

Columns of 11 cm length and 2.5 cm diameter were loaded with 49.0-57.5 g of soil to reach a bulk density of $1.4 \pm 0.1$ g cm$^{-3}$. The columns were loaded (from bottom to top) with 1.5 cm of decarbonized and combusted quartz sand, 7 cm soil, 1 cm of a soil-PyOM mixture and 1 cm quartz sand. The soil-PyOM mixture contained 0.5 g of PyOM and 4.5 g of soil for each column, which is equivalent to 11 t PyOM ha$^{-1}$ input on the soil surface. Control columns without addition of PyOM were packed with 8 cm of soil. For each soil, four replicates with addition of fresh and oxidized PyOM and controls were used. The columns were packed dry and saturated from the bottom using a 0.01 M CaCl$_2$ solution with a flow of 0.1-0.2 ml min$^{-1}$. The percolation was conducted with 0.01 M CaCl$_2$ from the top using Mariott's bottles for a constant pressure head and an adjusted flow of 2 ml min$^{-1}$. In total, 8,640 ml (on average $590 \pm 7$ soil pore volumes) were percolated through each column over five days. Subsamples of 200 ml were sequentially sampled after a percolation of 480, 1,440, 3,840, 6,240 and 8,640 ml. This is equivalent to 1,000, 3,000, 8,000, 13,000 and 18,000 l m$^{-2}$. The percolation was stopped for 3-5 h (flow interruption) during the sampling, but the columns were continuously saturated. The experiment was conducted at room temperature and the columns were protected from light to avoid photo-degradation. The percolated samples were not filtered to reduce the risk



of sample cross-contamination or losses. Therefore, the percolates contained the total mobile fraction of PyOM and nSOC including colloidal and dissolved forms.

## 2.4 Soils and percolate sample preparation and analyses

Amorphous Fe(o) and Al(o) were extracted by oxalate extraction according to McKeague et al. (1971) and measured using atomic absorption spectroscopy (ContrAA 700, Analytik Jena, Jena, Germany). Soil texture was determined after
oxidizing the organic material with $H_2O_2$, after which samples were wet sieved <63 mm and the silt/clay fraction further quantified by a Sedimat 4-12 (UGT, Müncheberg, Germany).

At each sequential sampling time, the pH and EC of the percolates was measured (914 pH/Conductometer, Metrohm, Herisau, Switzerland). The pH and EC of bulk soils was measured using a soil-solution ratio of 1:2.5 in a 0.01 M $CaCl_2$ solution after shaking and settling for one hour.

The percolate samples were stored at 4°C for a maximum of three weeks. If longer storage was required, the samples were stored frozen. All liquid samples were freeze dried and weighed prior to further analysis.

The soil columns were sampled after drainage form the bottom to the top to avoid any cross-contamination with labelled material from soil-PyOM layer. Three soil layers of 2.3 cm below the soil-PyOM layer were sampled and dried at 40°C over night, corresponding to 0-2.3, 2.3-4.6 and 4.6-7.0 cm depth layers.

Density fractionation was conducted with 5 g of the first 0-2.3 cm below the soil-PyOM layer and bulk soil samples. A sodium polygunstate (SPT) solution adjusted to a density of 1.8 g $cm^{-3}$, as recommended by Lavallee et al. (2019), was used to separate the fPOM after shaking and settling for one hour. The floating fPOM was decanted after centrifugation (30 min at 4000 rpm) and vacuum filtered using a glass fiber filter. The fPOM on the filter was rinsed with deionized water to remove SPT. The remaining heavier MAOM was rinsed with deionized water and centrifuged for three times to remove SPT. No
further fractions were acquired and the commonly used ultrasonification, to separate occluded particulate organic matter, was avoided to reduce the risk of physical breakdown of PyOM particles and potential shift between density fractions. All samples were dried at 40°C over night.

Soil and density fraction samples were milled for further measurement. If the sample mass of the fPOM fraction was too little, the samples were ground manually with a mortar and pestle to reduce a potential loss of sample material. The freeze-
dried percolate samples were homogenized by manual grinding. Bulk soil and bulk PyOM samples were dried at 40°C over night and milled. The total C and $\delta^{13}$C, relative to the international Vienna Pee Dee Belemnite (VPDB) standard, of all solid samples were measured using a dry combustion module cavity ring-down spectroscopy system (Picarro, Santa Clara, USA). Since our soils were carbonate free (pH<6), the total organic C (TOC) was equal to the measured total C.

## 2.5 Hydraulic properties of soil columns

Following the percolation experiment, breakthrough curves (BTC) were conducted in order to evaluate similar hydraulic properties and flow conditions. Afterwards the soils were sampled as described above. The BTC were performed





using a NaCl (1 g l$^{-1}$) solution as an inert tracer. The tracer solution was added for 25 minutes with a constant pressure head allowing an average flow of 1 ml min$^{-1}$. Afterwards, the percolation was continued with 0.01 M CaCl$_2$ for further 47 minutes. The EC of the percolated solution was measured in 12 fractions each sampled after six minutes (see Fig. S2 for all BTCs). The

percolated volume was added to the last percolate fraction.

The convective velocity $v$ [cm min$^{-1}$] and the diffusion coefficient $D$ [cm$^2$ min$^{-1}$] were estimated by using STANMOD (Version 2.08.1130; Šimůnek $et\ al.,$ 2003) to solve the deterministic equilibrium convection-diffusion equation. Due to high flow rates under saturated conditions, it can be assumed that diffusion of the tracer within the soil was negligible and the dispersivity $\lambda$ [cm] was calculated as the quotient of the $D$ and $v$ (Vanderborght and Vereecken, 2007). No significant

differences of dispersivity were found between the columns of the same soil: $0.29 \pm 0.04$ cm for the loamy topsoil, $0.19 \pm 0.04$ cm for the loamy subsoil, $0.13 \pm 0.01$ cm for the sandy topsoil and $0.14 \pm 0.02$ cm for the sandy subsoil (see Table S1 for all values). Therefore, the packing and the addition of the soil-PyOM layer did not lead to any significant trend for changes in the hydraulic properties of the columns and thus hydraulic parameter were excluded as further explanatory variables in this experiment.

**2.6 Calculations and statistics**

The recovery of $^{13}$C derived from the labelled PyOM was calculated with the atomic $^{13}$C fractions following the recommendations presented by Coplen (2011). The measured $\delta^{13}$C values were used to calculate the isotope-amount ratios $R(^{13}C/^{12}C)_{sample}$ of each sample, using an isotope-amount ration of 0.01118 for the VPDB standard. The atomic fraction of each sample $x(^{13}C)_{sample}$ was calculated following Eq. (1):

$$x(^{13}C)_{sample} = \frac{R(^{13}C/^{12}C)_{sample}}{1 - R(^{13}C/^{12}C)_{sample}}$$  (1)

The excess isotope-amount fraction of each sample $xE(^{13}C)_{sample}$ was calculated following Eq. (2):

$$xE(^{13}C)_{sample} = x(^{13}C)_{sample} - x(^{13}C)_{control}$$  (2)

where $x(^{13}C)_{control}$ is the atomic $^{13}$C fraction of the corresponding soil. Here, the mean of the control columns was used to calculate the excess isotope-amount fraction of the individual soil column depth after the percolation. The mean atomic $^{13}$C

fraction of all controls of the first percolate (1,000 l m$^{-2}$) was used to calculate the excess isotope-amount fraction of percolates from soil columns with addition of PyOM. The first percolates were observed to provide the most stable values for the atomic $^{13}$C fraction of the control due to higher C contents compared to later percolates with less total C. The recovery of $^{13}$C in [mg] was calculated following Eq. (3):

$$m_{recovery_{13C}} = \frac{xE(^{13}C)_{sample}}{xE(^{13}C)_{soil-PyOM}} \times m_{sample}$$  (3)

where $xE(^{13}C)_{soil-PyOM}$ is the excess isotope-amount fraction of the corresponding soil-PyOM mixture and $m_{sample}$ is the total mass of C measured in the sample in [mg]. This recovery calculation allowed to distinguish between three separate C



pools: (1) TOC, (2) the labelled PyOM-C and (3) nSOC. On average, the total recovery of TOC in the soil and percolates was $91.6 \pm 1.0$ % and $91.4 \pm 2.7$ % % of added PyOM-C. The recoveries were normalized to 100 % for further comparison.

Significant differences in PyOM-C between fresh and oxidized PyOM treatments were tested with a t-test. To test significant differences of nSOC in soils and percolates between controls and treatments with addition of PyOM, analysis of variance (ANOVA) was applied and p-values were computed with the post-hoc Tukey's 'Honest Significance Difference' of means method on a 95 % family-wise confidence level. The statistical analyses were performed using R version 4.0.0 (R Core Team, 2020). The standard error (error of the mean) of four replicates is presented for all data and error propagation was applied for cumulative nSOC and PyOM-C fluxes. Due to a potential sample contamination after the experiment, one control replicate of the sandy subsoil was excluded from the analysis. One replicate of the loamy subsoil with addition of oxidized PyOM was excluded from further analysis due to an unsteady flow during the percolation.

## 3 Results

### 3.1 Mobility of fresh and oxidized PyOM-C

The percolation and export of oxidized PyOM-C from the columns was higher compared to fresh PyOM-C for all soils over the whole percolations (Fig. 2). After a total percolation of $18,000 \, l \, m^{-2}$, $0.70 \pm 0.15$ % of oxidized and $0.24 \pm 0.03$ % of fresh PyOM-C percolated from the sandy topsoil (p<0.01). From the sandy subsoil, $0.64 \pm 0.08$ % of oxidized and $0.26 \pm 0.07$ % of fresh PyOM-C were leached (p<0.01). The loamy topsoil showed a percolation of $0.40 \pm 0.25$ % of oxidized and $0.18 \pm 0.03$ % of fresh PyOM (p=0.16). Significantly more oxidized PyOM-C leached from the loamy subsoil, with $0.40 \pm 0.06$ % of oxidized PyOM-C compared to $0.17 \pm 0.04$ % of fresh PyOM-C (p<0.01). The total amount of percolated oxidized and fresh PyOM-C did not differ significantly between topsoil and subsoil for the loamy or the sandy soil. Between the two soils, the export of oxidized PyOM-C was significantly increased (p<0.01) for the sandy topsoil and subsoil compared to the loamy soil while the export of fresh PyOM-C was not significantly different (p=0.38).

The first flush of $1,000 \, l \, m^{-2}$ caused the highest export of PyOM-C from the soil columns (Table S2) and contributed to 80.4-84.3 % of total percolated PyOM-C from the sandy soil, and to 50.6-79.8 % of the total percolated PyOM-C from the loamy soil (Fig. 2). The first flush of PyOM-C was similar between the sandy topsoil and subsoil (2,862.9-1,114.7 µg PyOM-C $l^{-1}$) and on-going over the whole percolation time. For the loamy topsoil, the first flush indicated higher PyOM-C concentrations (596.4-1,411.4 µg PyOM-C $l^{-1}$) compared to the subsoil (347.5-724.2 µg PyOM-C $l^{-1}$), but the concentrations decreased to non-detectable levels for the last percolation stage of $18,000 \, l \, m^2$ from the loamy topsoil and were on-going for the subsoil.

### 3.2 pH and electrical conductivity of percolates

The initial pH of the percolates increased with addition of PyOM for all soils (Table S2, Fig. S3). As an average of all percolates, the pH increased significantly by $0.17 \pm 0.02$ units in the percolates of the sandy topsoil. This effect was larger





in the sandy subsoils with a significant increase by 0.31-0.46 units. For the loamy topsoil, the first flush ($1,000\ l\ m^{-2}$) showed significantly increased pH values by 0.46-0.76 units with addition of PyOM, but thereafter approached the pH of the percolates from the control. Changes in pH were less dominant in the loamy subsoil but the pH increased continuously over the whole percolation.


The EC increased significantly with addition of PyOM ($p<0.01$) in the first percolates compared to the control for all soils (Table S2, Fig. S4). With further percolation, the EC equilibrated to the background value of the percolate solution ($2.20\ mS\ cm^{-1}$ of $0.01\ M\ CaCl_2$) for all soils with and without addition of PyOM.

### 3.3 Percolated nSOC


The addition of PyOM significantly increased the total percolated nSOC from the sandy topsoil and subsoil compared to the control, but did not change the total nSOC export from the loamy soil (Fig. 3). The addition of fresh PyOM significantly increased the total nSOC percolation compared to the oxidized PyOM in the sandy topsoil and subsoil ($p=0.01$). In total, $3.5 \pm 0.6\ \%$ ($0.73 \pm 0.05\ g\ nSOC\ kg^{-1}\ soil$) of total initial nSOC leached with fresh PyOM and $2.5 \pm 0.2\ \%$ ($0.53 \pm 0.02\ g\ nSOCC\ kg^{-1}\ soil$) of total nSOC with addition of oxidized PyOM from the sandy topsoil. This was significantly more to than the control with an export of $1.7 \pm 0.1\ \%$ of the total initial nSOC ($p<0.01$). From the sandy subsoil, $4.9 \pm 0.4\ \%$ of the total nSOC percolated from the control while significantly more nSOC percolated with fresh ($9.9 \pm 1.1\ \%$; $0.26 \pm 0.01\ g\ nSOCC\ kg^{-1}\ soil$) and oxidized ($7.8 \pm 0.5\ \%$; $0.21 \pm 0.01\ g\ nSOCC\ kg^{-1}\ soil$) PyOM ($p<0.01$).


Between the loamy and sandy topsoils, the total relative nSOC percolation did not differ significantly for the controls ($p=0.71$) and with addition of oxidized PyOM ($p=0.86$). But the addition of fresh PyOM resulted in a higher nSOC percolation in the sandy topsoil compared to the loamy topsoil ($p=0.01$). For the subsoils, the total nSOC percolation was significantly higher in sandy subsoil compared to the loamy subsoil with fresh PyOM ($<0.01$) and oxidized PyOM ($p<0.01$) but not for the controls ($p=0.28$).


The first flush ($1,000\ l\ m^{-2}$) showed the highest nSOC concentrations in the percolates (Table S2). From the sandy topsoil, $45.54 \pm 7.69\ mg\ nSOC\ l^{-1}$ ($p<0.01$) with fresh and $28.16 \pm 2.09\ mg\ nSOC\ l^{-1}$ ($p=0.05$) were leached with addition of oxidized PyOM, whereas significantly less nSOC was leached from the control ($12.12 \pm 0.24\ mg\ nSOC\ l^{-1}$). The nSOC concentrations in the percolates were lower for the sandy subsoil ($13.31$-$3.28\ mg\ nSOC\ l^{-1}$) compared to the topsoil, but higher for the last percolate indicating an on-going nSOC mobilization. The nSOC percolated from the loamy topsoil in the first flush was $9.65 \pm 1.07\ mg\ nSOC\ l^{-1}$ for the control and significantly increased to $21.14 \pm 1.96\ mg\ l^{-1}$ ($p=0.01$) and $15.89 \pm 2.08\ mg\ l^{-1}$ ($p=0.16$) with addition of fresh and oxidized PyOM, respectively. The loamy subsoil showed lower leached nSOC concentrations in the percolates with $7.86 \pm 0.49\ mg\ nSOC\ l^{-1}$ compared to the topsoil with addition of fresh ($12.23 \pm 1.55\ mg\ nSOC\ l^{-1}$; $p=0.07$) and with oxidized PyOM ($10.92 \pm 0.79\ mg\ nSOC\ l^{-1}$; $p=0.07$).





### 3.4 Changes in fresh and oxidized PyOM-C and nSOC in soil columns

The soil-PyOM layers contained on average of all soils 90-96 % of the total initially added fresh and 89-96 % of the
total initially added oxidized PyOM-C after the percolation. The first 0-2.3 cm below the soil-PyOM layer contained the largest
proportions of mobilized PyOM with no differences between oxidized and fresh PyOM (Fig. 4). The recovered PyOM-C from
this layer ranged between 3.5-9.7 % (0.84-1.50 g PyOM-C kg$^{-1}$ soil) and 3.6-10.1 % (0.52-1.06 g PyOM-C kg$^{-1}$ soil) of added
PyOM-C in the sandy and loamy soil, respectively. With greater soil depth, the recoveries decreased to <1 %. In the soil layers
at 2.3-4.6 cm and 4.6-7.0 cm below the soil-PyOM layer, only 0.01-0.13 % and 0.05-0.17 % of added fresh and oxidized
PyOM-C were recovered, respectively. The recovery of oxidized PyOM-C in these layers was mostly significantly higher
compared to the fresh PyOM-C.

Between topsoil and subsoil, the first layer below the soil-PyOM layer (0-2.3 cm) showed no significant differences
in the recovery of PyOM-C for both, sandy and loamy soil. The sandy subsoil indicated higher recoveries of oxidized PyOM-
C (p=0.12) compared to the topsoil in 2.3-4.6 cm depth but not for fresh PyOM-C (p=0.68). The lowest sandy subsoil layer,
however, showed significant higher recoveries of oxidized (p<0.05) and fresh PyOM-C (p<0.01) compared to the topsoil. The
loamy subsoil showed significantly higher fresh (p=0.05) and oxidized PyOM-C (p<0.01) recoveries in 2.3-4.6 cm depth
compared to the topsoil. For the deeper layer (4.6-7.0 cm), the recovery of oxidized (p<0.05) and fresh PyOM-C (p<0.01) were
also higher in the subsoil than in the topsoil. The relative recovery of PyOM-C did not differ significantly between the loamy
and sandy soils for the same depths.

The total nSOC loss from the soil columns significantly increased with PyOM for the sandy soil, but not for the loamy
soil (Fig. 5). The nSOC contents decreased by 1.8 ± 0.2 % (3.9 ± 2.6 g nSOC kg$^{-1}$ soil) with fresh PyOM and by 0.8 ± 0.1 %
(1.8± 1.3 g nSOC kg$^{-1}$ soil) with addition of oxidized PyOM in the sandy topsoil (p<0.01). In the sandy subsoil, the PyOM
resulted in 4.8 ± 0.5 % (0.4± 0.2 g nSOC kg$^{-1}$ soil) and 2.6 ± 0.3 % (0.3± 0.2 g nSOC kg$^{-1}$ soil) lower nSOC contents after
percolation with fresh and oxidized PyOM, respectively (p<0.01). The losses of nSOC were significantly larger with fresh
PyOM than with oxidized PyOM in the sandy topsoil and subsoil (p=0.01).

### 3.5 Density fractionation of 0-2.3 cm below soil-PyOM layer

The density fractionation of the first 0-2.3 cm below the soil-PyOM layer revealed that large proportions of the
PyOM-C remained in the light fPOM fraction in the sandy soil (Fig. 6). In the sandy topsoil, 93.3± 0.9 % of the PyOM-C was
found in the fPOM fraction. The same fraction contributed to 86.3 ± 1.7 % of the total PyOM-C in the first sandy subsoil depth
below the soil-PyOM layer (0-2.3 cm). In the loamy soil, 40.2 ± 1.8 % of the PyOM-C was associated with the MAOM fraction
in the topsoil and subsoil, given as an average of fresh and oxidized PyOM. In general, the proportions of PyOM-C found in
the two fractions did not differ significantly between the fresh and oxidized PyOM and between the loamy topsoil and subsoil.
The sandy subsoil indicated significantly more oxidized PyOM-C in the MAOM fraction compared to the topsoil (p=0.05).



## 4 Discussion

**4.1 Mobility of PyOM**

At the end of the experiment, 3.8-10.8 % of the added PyOM moved vertically from its initial location. This includes PyOM recovered in seven centimeters of soil below the soil-PyOM application layer and exported PyOM in the percolates (Fig. 7). Large parts of the mobilized PyOM were translocated to the first 0-2.3 cm below the soil-PyOM layer (3.5-10.1 % of added PyOM-C; Fig. 4). The total recovery of PyOM in this layer did not differ significantly between the sandy and loamy

soil. This indicates a potential accumulation close to the transition of the soil-PyOM layer and the upper 0-2.3 cm regardless of the soil texture. The PyOM-C recovered in deeper soil depth of 2.3-7.0 cm and in the percolates was most likely derived from dissolved and colloidal PyOM. Dissolved and colloidal PyOM is reported to be a major mobile fraction in soils (Wagner et al., 2017, 2018).

Abiven et al. (2011) reported that 0.31-0.42 wt% of PyOM were mobilized in a batch experiment without any soil

addition as dissolved (<0.45 µm) and colloidal (0.45-5 µm) forms. In field incubation experiments (one to two years), a small proportion of 0.041-0.004 % of initially added PyOM was reported to be mobilized vertically to >15 cm depth in dissolved forms (Maestrini et al., 2014; Major et al., 2010). Hilscher and Knicker, (2011) reported that 2.3 % of added PyOM migrated to 5 cm depth and 0.4 % were exported from the soil (>8 cm depth) in a one-year incubation experiment. We identified that 0.17-0.70 % of the added PyOM-C were exported from the soil columns (>7 cm; Fig. 2). This is in accordance to the relatively

limited mobility of PyOM observed under experimental and field conditions compared to non-pyrolyzed OM.

Don and Kalbitz (2005) reported that fresh and 12 month in-situ incubated litter from a variety of temperate forest trees (e.g. sycamore maple, mountain ash, beech, spruce and pine litter) can release 0.3-6.5 % of water extractable DOC. Liebmann et al. (2020) applied highly [13]C labeled beech litter on the surface of a temperate forest floor and reported that after 22 months 11.2 % of the applied litter migrated within a depth of 0-140 cm, but 87 % of this mobilized litter fraction was

found in the upper 5 cm, indicating a minor importance of aboveground litter DOC input in deeper soils. However, the mobilized PyOM is most likely less controlled by microbial decomposition due to the high stability of PyOM (Kuzyakov et al., 2014; Singh et al., 2012), allowing a potential deeper migration. It needs to be noticed that we only included one type of PyOM (ryegrass derived and produced at 450°C) which constrains general assumptions.

Under field conditions, pedoturbation (e.g. due to swelling and shrinking of clay rich soils) and bioturbation would

potentially promote the vertical translocation of PyOM (Hobley, 2019; Rumpel et al., 2015). Physical fragmentation and break-down of PyOM during environmental aging reduces the particle size, which can increase the vertical mobility of PyOM due to decreasing friction of smaller particles during transport and the generation of colloids (Hobley, 2019; Pignatello et al., 2015; Spokas et al., 2014). These colloids tend to become more mobile with ageing, decreasing particle size, increasing pH, and in the presence of OM, whereas their mobility tends to decrease in the presence of clay minerals, hydrophobic contaminants, and

high ionic strengths due to aggregation. (Castan et al., 2019; Sigmund et al., 2018). Fragmentation further increases the relative




specific surface area of PyOM and may promote stabilization and thus its retention in soils due to increased physico-chemical interaction with the mineral phase (Singh et al., 2012; Xiao and Pignatello, 2015).

## 4.2 Dynamic of mobilized PyOM

The first flush contributed to the highest export of PyOM from the soil columns and the mobilized amounts decreased
with the percolation for all soils (Fig. 2 and Table S2). We hypothesized a continuous export of PyOM-C and decrease with time which can be confirmed with our experiment. Our results clearly indicate that the first flush is a major event of PyOM transport through soil and contributed to 80-84 % and 51-79 % of total exported PyOM from the sandy and loamy soil, respectively. This can be attributed to mobile PyOM fractions which are directly produced during the pyrolysis and easily mobilized with the initial flux (Hilscher et al., 2009). Therefore, the initial flux of PyOM may significantly contribute to its
total export from soils and its transition to aquatic systems under field conditions. Most field experiments and observation miss the initial PyOM flux (lateral and vertical) with the first rain event after a fire, resulting in a potential underestimation of the dissolved transport rates of PyOM (Santos et al., 2017). This may further cause an underestimation of the vertical PyOM transport from depositional landscape positions after a redistribution following an initial lateral mass transport (Rumpel et al., 2015).

The PyOM-C export was continuous for the sandy topsoil and subsoil indicating an on-going mobilization and migration through the coarse soil. Besides this large export of PyOM with the first flush, PyOM-C was not detectable in the last percolated fraction of the loamy topsoil. Therefore, large proportions of PyOM were able to migrate through the loamy topsoil with the first flush, but with decreasing PyOM-C concentration, the retention and most likely the sorption to the mineral phase and OM increased. In comparison, the loamy subsoil retained larger proportions of the PyOM mobilized with the first
flush (flush of 51-58 % of the total exported PyOM) but continuously released the retained PyOM into the solution with further percolation.

The on-going PyOM export from soils in our experiment could potentially explain the steady, less seasonally affected and from fire history decoupled flux of soil derived PyOM to rivers as observed in the field (Bao et al., 2019; Dittmar et al., 2012a, 2012b; Santos et al., 2017; Wagner et al., 2018). In our experiment, the proportion of percolated PyOM-C of the total
percolated C ranged between 2.4-17.2 % for the first flush and between 0.2-2.8 % for the last percolates, as an average of all soils (Table S2). Artificially produced PyOM is mostly more stable than naturally produced PyOM, which challenges the use of one type as a proxy for the other (Santín et al., 2017). The proportions observed here, however, are in line with globally reported dissolved black carbon proportions on total riverine (3-15 %) and marine (0.1-2.9 %) DOC (Coppola and Druffel, 2016; Jaffé et al., 2013; Jones et al., 2020; Wagner et al., 2018). However, we did not investigate degradation (biotic or abiotic)
of mobilized PyOM and nSOC after the export from the soil, which would increase the proportion of PyOM-C on the total DOC.



### 4.3 Effect of oxidation on PyOM mobility and reactivity

The oxidation significantly increased the PyOM mobility and resulted in 2-7 times higher PyOM-C recoveries in the soil in 2.3-7.0 cm (Fig. 4) and 2-3 times higher losses through percolation (Fig. 2) compared to fresh PyOM (Fig. 7). This

confirms our second hypothesis that oxidation (aging) is enhancing the mobility of PyOM. The similar quantities of fresh PyOM-C found in the percolates from the sandy and loamy soil indicates that the mobility of fresh PyOM is most likely only partially influenced by soil texture and rather controlled by the total abundance of mobile compounds originated directly from the pyrolysis. For the oxidized PyOM, however, the exported quantities of PyOM-C revealed a nearly twofold higher mobility in the sandy soil than in the loamy soil. This indicates a higher mobility of PyOM with aging in coarse textured soils and a

potentially higher retention of the oxidized PyOM in soils with a finer texture.

In a batch experiment, Abiven et al. (2011) reported an increase of water extractable PyOM fraction by 40-50 times after 10 years of natural aging compared to recent PyOM and a higher aromaticity of the solubilized aged PyOM. This, however, was estimated without any soil interaction of the mobile fraction. Hockaday et al. (2006) provided indirect evidence that PyOM in a fire-affected watershed is mainly derived from PyOM previously aged in soils and that it is undergoing a

fractionation during its migration through the soil depending on its initial aromaticity. In a recent study, Braun et al. (2020) reported a decreasing aromaticity of water extractable PyOM extracted from agricultural soils compared to the bulk PyOM and no changes after three years aging under field conditions. Contrastingly, Wagner et al. (2017) reported an increased aromaticity of PyOM exported from soils with aging (>100 years) compared to fresh bulk PyOM. The authors, however, reported that the oxidation alone could not explain the reported higher mobility of PyOM with aging. This highlights that

oxidation is not only increasing its mobility but also its reactivity in soil during aging and its transport.

The reactivity of PyOM in soils clearly increased with oxidation in our experiment and caused a higher retention of oxidized PyOM in 2.3-7.0 cm soil depth below the soil-PyOM layer in both soils (Fig. 4). This increased retention can be associated with enhanced interaction of the oxidized PyOM surfaces with the soil mineral phase and the native soil OM. Scanning electron microscopy showed a preferential interaction of partially oxidized PyOM and the soil mineral phase

(Brodowski et al., 2005). The increased PyOM reactivity and mobility with oxidation can be attributed to the higher abundance of functional groups containing hydrogen and oxygen (e.g. carboxyl groups) on the oxidized surface (Fig. 1). This oxidation consequently increased the polarity which enhances its water solubility and its interaction with the mineral phase (Cheng et al., 2008; Fang et al., 2014; Pignatello et al., 2015; Zhao and Zhou, 2019; Zimmerman, 2010).

Our results show that the long-term fate of PyOM in soils is highly controlled by its degree of oxidation and thus will

change with aging. However, aging will not only increase the PyOM mobility and solubilization but also its reactivity and thus interaction with the soil mineral phase and OM and thus its long-term sequestration. Therefore, future studies should include the effect of aging for more than a few years under field conditions.



## 4.4 PyOM retention in soil

The majority of the mobilized PyOM was retained in the soil (92.8±1.0 %). This indicates the importance of PyOM
retention in soils during its migration and thus confirms our third hypothesis. The density fractionation of the first soil layer
(0-2.3 cm) below the soil-PyOM layer revealed that most of the translocated PyOM remained as fPOM with 85.6-94.5 % and
56.9-61.4 % in the sandy and loamy soil, respectively (Fig. 6). More than one third of the PyOM (40.2 ±1.8 %) in the loamy
topsoil and subsoil, 0-2.3 cm below the soil-PyOM layer, however, was found to be association with the mineral phase. This
is surprising considering the relatively short interaction time of PyOM and the soil mineral surface and soil aggregation in a
saturated column experiment.

The soils in our experiment showed large differences in the contents of amorphous Fe(o) and Al(o) and in texture
with 13-15 % clay and 22-23 % silt in the loamy soil compared to <12 % silt and clay combined in the sandy soil (Table 1).
The aromatic compounds of the PyOM can directly interact with the edge functional groups of clay (Joseph et al., 2010;
Lehmann et al., 2007; Pignatello et al., 2015). Brodowski et al. (2006) found up to 24 % of PyOM in forest soils to be occluded
in aggregates and concluded that PyOM can act as a binding agent for the aggregate formation. In a field incubation experiment
of ten months, Singh et al. (2014) recovered >25 % of the added PyOM in the occluded fraction in temperate forest soils and
Vasilyeva et al. (2011) found up to 70 % of PyOM associated to the mineral fraction after 55 years of bare fallow in Chernozem.
Here, the MAOM fractions contain both, PyOM which is in direct association with the mineral phase and occluded in
aggregates. Our results show that the PyOM-mineral interaction and occlusions can occur very quickly in soils with reasonable
clay and silt content and may be predominately controlled by the physical and chemical interaction rather than biological
processes, since these were neglectable in our experiment. This quick mineral interaction will control the long-term stability
of PyOM in soils and requires more research under unsaturated and field conditions.

The higher proportion of PyOM recovered as fPOM in the sandy soil indicate that the PyOM-mineral interaction was
limited which is in agreement with the general higher proportion of TOC in fPOM fraction (41-73 %) and high sand contents
(>88 %; Table 1). In coarse soils, Fe- and Al-(hydr)oxides are considered to interact and stabilize PyOM and SOC due to its
great affinity to OM (Brodowski et al., 2005; Pignatello et al., 2015; Wiesmeier et al., 2019). Slightly more fresh PyOM
(p=0.24) and significantly more oxidized PyOM (p=0.05) was recovered in the MAOM fraction in the sandy subsoil compared
to the topsoil. This indicates that the sandy subsoil had a higher potential to retain PyOM due to increased mineral interaction
compared to the topsoil.

Subsoils are unsaturated in OM due to little inputs, such as DOC from upper soil horizons, root derived and already
microbially processed OM, but large mineral surfaces (Kaiser and Kalbitz, 2012; Lützow et al., 2006). Therefore, the
probability of mobilized PyOM to interact with the mineral phase is higher in the subsoil, resulting in a higher retention
compared to topsoils where OM is already occupying sorption sites on the mineral surfaces. The subsoils of the loamy soil
contained significantly more PyOM at 2.3-7.0 cm depth below the soil-PyOM layer compared to the topsoil (Fig. 6 and 7).
This was also the case for the last soil layer (4.6-7.0 cm) of the sandy subsoil and topsoil. Therefore, the loamy and sandy



subsoils had a higher PyOM retention potential than topsoils, where the nSOC contents were five to ten times higher. It needs to be noticed, that the PyOM retained in this depth represented only <1 % of the added PyOM. In addition, we did not include unsaturated flow conditions which would resemble the mobility and retention under field conditions.

The higher retention of PyOM in the subsoils can explain its accumulation in greater soil depth which is found under field conditions (Brodowski et al., 2007; Soucémarianadin et al., 2019). We showed that PyOM can be continuously re-mobilized from subsoils by percolating water and thus can potentially be exported to deeper soil depth or to the groundwater which would finally result in the export from the terrestrial to the aquatic system such as rivers. Furthermore, it can be considered that large proportions of the mobilized PyOM will not be affected by microbial degradation to the same extent than non-pyrogenic SOC and DOC during the percolation through the soil (Don and Kalbitz, 2005; Kuzyakov et al., 2014; Tipping et al., 2012). This would result in increasing proportions of PyOM on the total subsoil OM.

## 4.5 Effects of PyOM on nSOC mobility

The addition of PyOM significantly enhanced the total nSOC export in the sandy topsoil and subsoil by 48-270 % which is equivalent to an additional loss of 0.07-0.37 g nSOC $kg^{-1}$ soil (Fig. 3, Fig. 5 and Table S2). The nSOC export from the loamy soil was significantly increased compared to the control for the first flush (1,000 l $m^{-2}$) by 56-105 % (0.04-0.10 g C $kg^{-1}$ soil), but negligible over the whole percolation. Therefore, our fourth hypothesis can be partially accepted and the effect of PyOM on nSOC mobility was clearly controlled by soil texture and properties of the soil and PyOM.

With PyOM, the pH of the percolates increased in all soils except of the loamy subsoil which can further enhance the nSOC mobility (Table S2, Fig. S3). Increases in pH can be associated to a liming effect due to carbonates which are formed during the pyrolysis (Smebye et al., 2016). The pH significantly controls the mobility of DOC (Kaiser and Guggenberger, 2000; Kalbitz et al., 2000) and an increase by 0.5 pH units was shown to enhance the DOC export by 50 % (Tipping and Woof, 1990). We found increases in pH by 0.17-0.76 units (Table S2, Fig. S3). The liming effect lasted for the sandy soil over the whole percolation, which also showed a continuously higher export of nSOC with PyOM compared to the control. Therefore, the increasing pH in the sandy soil, with low initial pH values of 3.4-4.1, may have caused a re-mobilization of adsorbed nSOC derived DOC. The fresh PyOM had a significantly higher liming effect as indicated in the first flush of the sandy topsoil, and the loamy topsoil and subsoil, while the oxidized PyOM caused a higher EC in the first flush (Table S2, Fig. S4). This can be attributed to a decarbonatization of PyOM with oxidation, which was also observed with the MIR analysis (Fig. 1). Therefore, a liming effect of PyOM on the nSOC mobility will potentially decrease with aging but is substantial in coarse soils.

Barnes et al. (2014) reported that wood-derived PyOM (mesquite biochar, 400°C, 4 h) resulted in higher DOC fluxes from organic poor sandy soil in a soil column experiment (filled with soil-PyOM mixtures containing 10 % wt% PyOM), but attributed this increase to PyOM derived C and not soil-derived-C. The authors found no increase in DOC flux from organic- and clay-rich soil but identified an increase in the aromaticity of the DOC and the authors identified that leachable PyOM fractions were lost but soil-derived C was retained with PyOM. The authors concluded that the net increase in DOC export from soils with moderate amounts of clay, silt and OM is limited. Major et al. (2010) reported that wood-derived PyOM



amendment (mango wood biochar, 400-600°C, 48 h) in a savanna soil (sandy clay loam) increased the flux of non-PyOM

derived particulate organic carbon and DOC by 2.3-4.1 times after two years under field condition. However, the authors

attributed these increases to a higher belowground net primary production and a corresponding increase in OM input with

PyOM. Jones et al. (2012) found no change in the DOC flux after PyOM (wood biochar, 450°C, 48 h) from a sandy clay loam

in a three-year field trial. By tracing the highly labelled PyOM-C, our results confirm that PyOM may have a limited effect on

the nSOC mobility in fine-textured loamy soils over the long-term since we could not identify significant differences in the

cumulative nSOC loss from the soil columns with fresh and oxidized PyOM. However, the first flush of dissolved PyOM

enhanced the loss of nSOC significantly in the loamy topsoil and subsoil. It is likely that this mobilized nSOC would be

adsorbed again during its further transport through the loamy soil (>7 cm).

   Dissolved organic matter mobility in sandy soils is mainly controlled by adsorption on Fe-/Al-(hydr)oxides. High

molecular weight compounds with aromatic structures are reported to have a higher sorption affinity and can desorb less

adsorptive compounds from soil mineral phases (Coward et al., 2019; Eusterhues et al., 2011; Kaiser and Kalbitz, 2012; Kalbitz

et al., 2000). This is also shown by a fractionation of DOC migration through the soil (Oren and Chefetz, 2012) Recently,

Zhang et al. (2020) identified a higher sorption affinity of dissolved organic matter derived from maize straw PyOM by

hydrophobic partition, H-bonding and electrostatic interactions compared to non-pyrolyzed dissolved organic matter.

Therefore, PyOM caused a re-mobilization of nSOC by desprobing it from complexes of the mineral phase and pre-existing

OM. Due to a higher abundance of less oxidized highly aromatic compounds from fresh PyOM compared to oxidized PyOM,

the desorption of nSOC was more dominant with fresh PyOM, while oxidized PyOM adsorbed mainly on free mineral surfaces

due to its higher reactivity.

   The mobilization effect of PyOM on nSOC was more pronounced in the sandy subsoil with 7.8-9.9 % of the initial nSOC

exported compared to 2.5-3.5 % of initial nSOC exported form the topsoil (Fig. 3). This supports the concept of continuous

sorption, microbial processing and desorption during the vertical OM migration and thus an accumulation of microbially

processed OM in subsoils as described by Kaiser and Kalbitz (2012) as the cascade concept. The binding of microbial

processed OM to the mineral surfaces is weaker than for plant derived compounds (high in aliphatic, aromatic and carboxylic

groups) and thus it is more easily mobilized by PyOM with a high sorption affinity towards the mineral surfaces. We found

that even a small fraction of mobilized PyOM may cause a significant mobilization of potentially labile nSOC. Even if this

effect decreased with oxidized PyOM, it indicates a long-term effect on nSOC mobility and influence of the C cycle of fire

effected soils.

## 5 Conclusion and implementation

The vertical mobility of PyOM was limited to only a small fraction (<11 %) migrating through the soil columns. Large

proportions of the mobilized PyOM were retained in the soil and accumulated mainly in particulate form close to the initial

layer within the first 0-2.3 cm. Less than 1 % of the added PyOM migrated to greater depth and was exported from the soil



(>7 cm). Furthermore, the majority of the exported PyOM was mobilized with the first flush. Oxidation and thus aging of PyOM, significantly increased its mobility and also its reactivity, resulting in an overall larger mobilization but also larger retention in the soil of oxidized PyOM compared to fresh PyOM. Both can be clearly ascribed to the oxidized PyOM surfaces with a higher abundance of oxygen and hydrogen containing functional groups. The migration of oxidized PyOM was further largely influenced by the soil texture resulting in a higher export from the sandy soil and higher retention in the loamy soil due to an increased association to the soil mineral phase.

Fresh and oxidized PyOM significantly increased the mobility of nSOC in the sandy soil and in the first flush of the loamy soil, but not over the whole experiment. This can be attributed to a higher sorption affinity of high molecular weight PyOM compounds to the mineral phase and thus a desorption of already sorbed and mineral associated nSOC, which will eventually be exposed to microbial degradation. The re-mobilization of nSOC was greater in the sandy subsoil compared to the topsoil, supporting the concept that subsoil OM is already microbially processed and its association to the mineral phase is weak (because of low availability of mineral surfaces): hence it can be easily desobed by younger OM with a higher sorption affinity.

Further research is needed to understand the fate of PyOM under unsaturated and field conditions. However, we identified that the vertical PyOM mobility is highly depending on soil properties and the degree of PyOM oxidation (age) which increases not only its mobility, but also reactivity in soils and influences its effect on nSOC.

**Authors contribution**

MS and SA designed the experiment. MS conducted the experiment, analyzed the data and wrote the manuscript. SA, SB, GS and KK provided input to the data discussion and the manuscript.

**Competing interests**

The authors declare that they have no conflict of interest.

**Acknowledgment**

We thank Esmail Taghizadeh and Thomas Keller (University of Zurich) for support with analyses and technical assistance. This study was funded by the Swiss National Science Foundation (project no. 200021_178768)

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




**Table 1: Soil texture, total organic carbon (TOC), $\delta^{13}$C, pH, electrical conductivity (EC), oxalate extractable Fe(o) and Al(o) and density fractions (free particulate organic matter, fPOM and mineral associated organic matter, MAOM) for the topsoil (0-10 cm depth) and subsoil (40-60 cm) of the loamy and sandy soil (± 1 SE).**

| | Soil depth | Texture[1] | | | TOC | $\delta^{13}$C | pH (CaCl$_2$) | EC (CaCl$_2$) | Fe(o) | Al(o) | Density fractions | |
| | [cm] | [%] | | | [g kg$^{-1}$ soil] | [‰] | [-] | [mS cm$^{-1}$] | [g kg$^{-1}$] | [g kg$^{-1}$] | [% of total SOC] | |
| | | Sand | Silt | Clay | | | | | | | fPOM | MAOM |
| Loamy soil | 0-10 | 65 | 22 | 13 | 20.14 (0.67) | -29.1 (0.2) | 5.3 (0.1) | 2.33 (0.01) | 1.85 (0.17) | 1.30 (0.17) | 17.2 (0.2) | 82.8 (0.2) |
| | 40-60 | 62 | 23 | 15 | 5.66 (0.18) | -28.9 (0.2) | 4.0 (0.1) | 2.35 (0.03) | 2.12 (0.10) | 1.99 (0.15) | 14.3 (1.0) | 85.7 (1.0) |
| Sandy soil | 0-10 | 88 | 5 | 7 | 21.91 (0.01) | -28.8 (0.1) | 3.4 (0.1) | 2.45 (0.01) | 1.01 (0.12) | 0.43 (0.03) | 73.2 (0.6) | 26.9 (0.6) |
| | 40-60 | 92 | 3 | 5 | 2.64 (0.08) | -30.1 (0.2) | 4.1 (0.1) | 2.34 (0.01) | 0.93 (0.04) | 0.83 (0.10) | 41.1 (0.8) | 58.9 (0.8) |

[1]according to WRB texture classes

**Table 2: Total C and $\delta^{13}$C of PyOM used for the corresponding soil column experiment (± 1 SE).**

| PyOM | Soil | Type | Total C [%] | $\delta^{13}$C [‰] |
|---|---|---|---|---|
| 1. | Loamy topsoil | Fresh | 40.6 (0.8) | 4135.7 (5.3) |
| | | Oxidized | 42.1 (1.5) | 4201.7 (5.2) |
| 2. | Loamy subsoil | Fresh | 35.7 (1.2) | 3471.6 (7.7) |
| | | Oxidized | 38.9 (2.1) | 3457.5 (18.5) |
| 3. | Sandy topsoil & subsoil | Fresh | 48.5 (1.8) | 2877.0 (5.7) |
| | | Oxidized | 49.4 (1.3) | 2942.6 (3.2) |





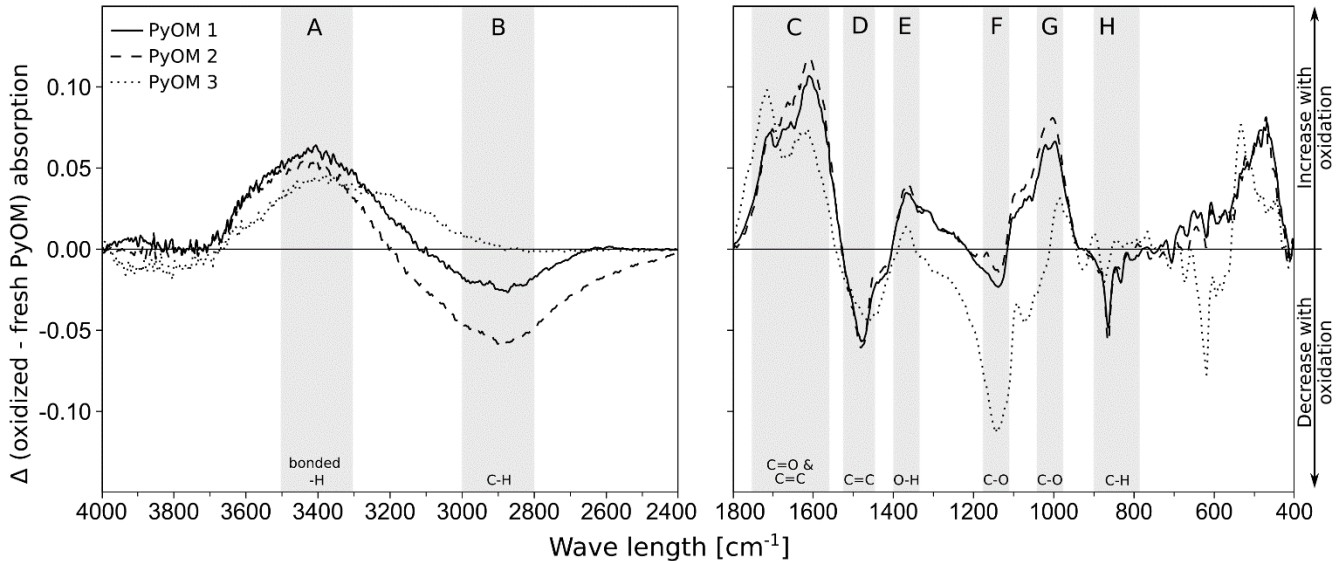

**Figure 1: Difference of oxidized and fresh PyOM DRIFT spectra for PyOM 1 (loamy topsoil), PyOM 2 (loamy subsoil) and PyOM**
**3 (sandy soil). Areas A-H indicate the main changes after oxidation with the corresponding absorption bands indicating increases in**
**A = 3500-3200 cm⁻¹, C-O bonds, hydroxyl groups and H₂O; C = 1730-1680 cm⁻¹, aromatic carbonyl/carboxyl C=O bonds and C=C**
**bonds (1590-1560 cm⁻¹ and 1620-1610 cm⁻¹); E = 1375 cm⁻¹, O-H bonds and G = 1060-1020 cm⁻¹, C-O bonds. Oxidation decreased**
**the absorption at bands B = 2980-2820 cm⁻¹, aliphatic C-H bonds; D = 1500 cm⁻¹ aromatic C=C bonds; F = 1280-1200 cm⁻¹, C-O and**
**H-O bonds and H = 880 and 805 cm⁻¹, aliphatic C-H bonds. Decreases at 1480 cm⁻¹ and 864 cm⁻¹ can be assigned to a decarbonation**
**with oxidation. See supplement (Fig. S1) for full spectra.**





**Figure 2: Cumulative leaching loss of percolated fresh and oxidized PyOM from sandy and loamy topsoil and subsoil. The p-values indicate the significance of differences between fresh and oxidized PyOM after a total percolation of 18,000 l m⁻². All values are shown with propagated SE.**

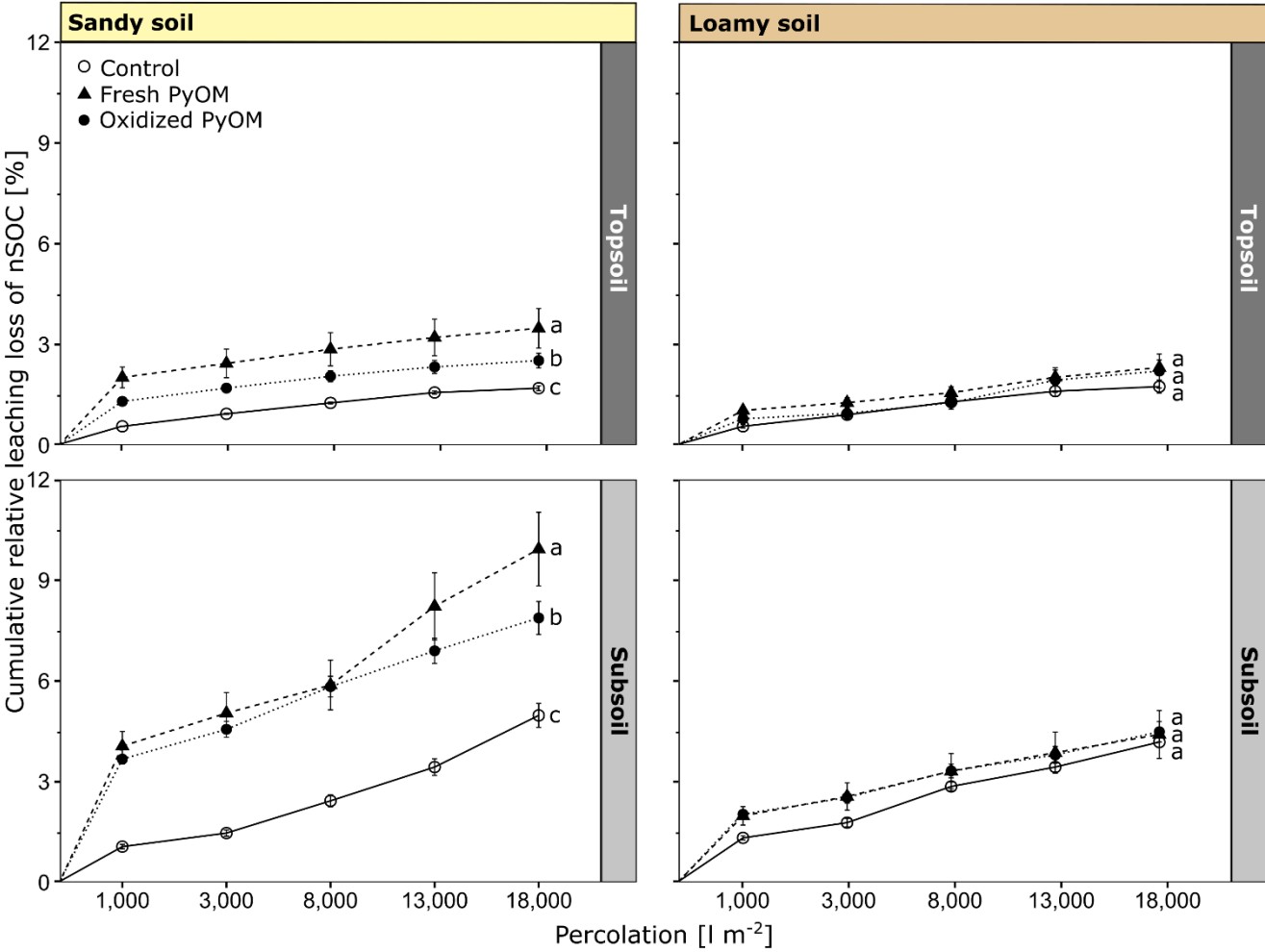

**Figure 3: Cumulative leaching loss of native soil organic carbon (nSOC) from sandy and loamy topsoil and subsoil for controls and**
**columns with addition of fresh and oxidized PyOM. The significant differences (p<0.05) of the total percolated nSOC after**
**18,000 l m$^{-2}$ are indicated by letters. All values are shown with propagated SE.**





**Figure 4: Recovery of PyOM-C in soil below the soil-PyOM layer of the sandy and loamy topsoil and subsoil (± 1 SE). The p-values indicate the significance of differences between fresh and oxidized PyOM at each depth.**





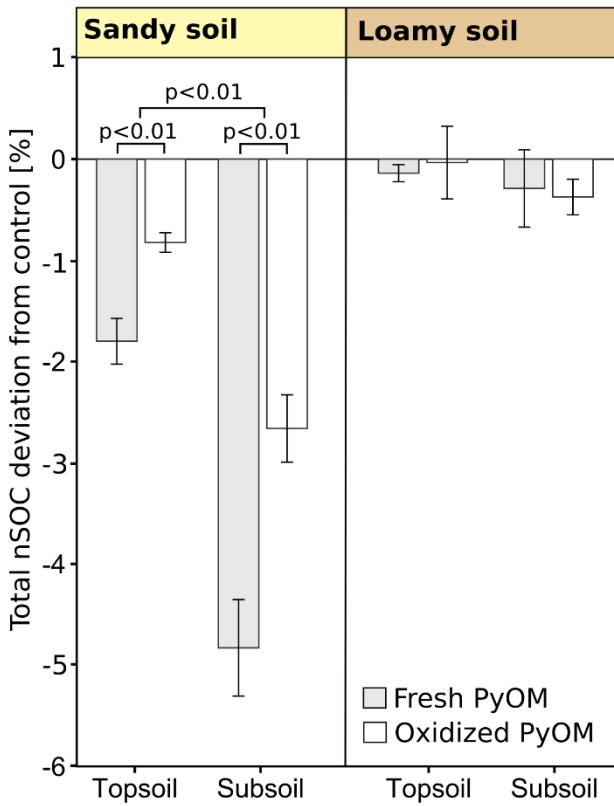

**Figure 5: Relative deviations of native soil organic carbon (nSOC) to control of the total soils in soil columns (over all depth) after**
**the percolation for sandy and loamy topsoil and subsoil with addition of fresh and oxidized PyOM (± 1 SE). Negative values indicate leaching losses. Significance of differences of fresh and oxidized PyOM and the significance of the deviation from the control is shown with p-values. No significant differences were found for the loamy soil.**






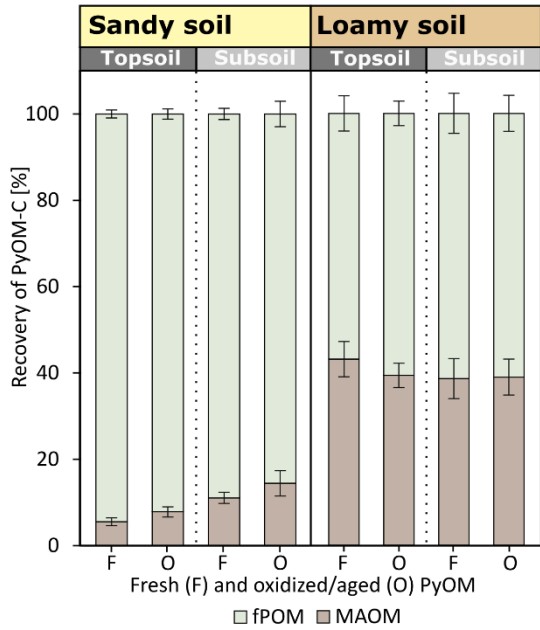

**Figure 6: Relative recovery of fresh (F) and oxidized (O) PyOM-C in fPOM and MAOM fractions in the first layer below the soil-PyOM layer (0-2.3 cm) of sandy and loamy topsoil and subsoil (± 1 SE).**




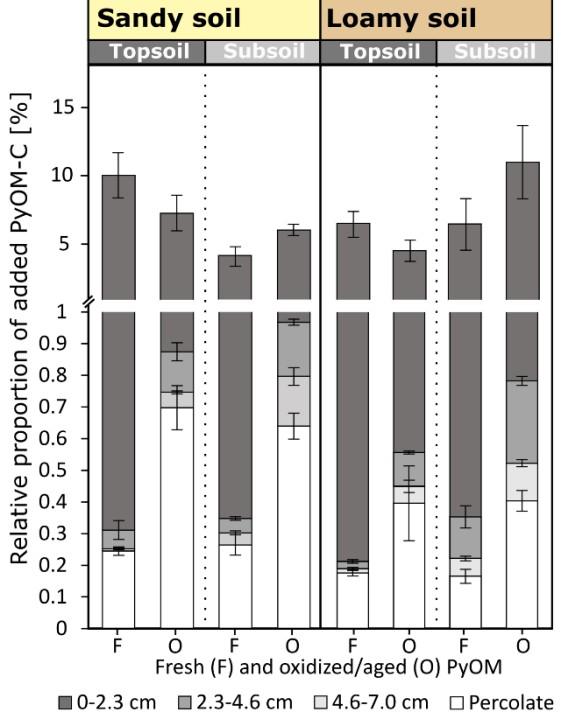

**Figure 7: Total mobilized fresh (F) and oxidized (O) PyOM and its relative proportion in the percolates after total percolation and in the soil column in 0-2.3, 2.3-4.6 and 4.6-7.0 cm below the soil-PyOM application layer for the sandy and loamy topsoil and subsoil (± 1 SE). The significances are presented in Figure 2 and 4.**