# Peer review of "Vertical mobility of pyrogenic organic matter in soils: A column experiment"

_Biogeosciences, 2020_

## Referee Comment (RC1) · Anonymous Referee #1 · 31 Aug 2020

General Comments

This manuscript provides information on the vertical mobility of pyrogenic organic matter in soils of contrasting nature. The study uses a column experiment where isotopically labelled char gets added to soils collected at different depths. The movement of both added PyOM and native soil is then traced through the soil column. The study is very relevant, well designed and impeccably executed. The MS is well written and provides a pleasant reading. It has been a few years since I did not enjoy reading a first submission as much as I have with this one. I have little doubt that this work will prove quite useful for researchers studying PyOM dynamics and ecosystem carbon cycling.

There are just a few aspects where I would appreciate that the authors provide further critical discussion to make this contribution an even more useful one. These are as

follows:

The PyOM used in this study derives from ryegrass. I can understand why such fast-growing precursor biomass was used to produce PyOM in this labelled experiment. However, it is unavoidable to think that the resultant pyrogenic material will be of a highly contrasting physico-chemical nature compared to those derived from woody vegetation. Therefore, these distinct characteristics may greatly affect the mobility of the various PyOM produced. As such, I recommend the authors to include a paragraph in the discussion showcasing the potential limitations and applicability of the results obtained in this study. The results obtained here may be directly applicable in agronomic studies using grass-derived biochar. However, the mobility of PyOM in charcoal generated during wildfires affecting woody vegetation might be different from that observed in grass-derived PyOM.

The existence of fluctuating levels of moisture in the soil is just natural. Please briefly include an statement about how soil drying and wetting events may cause the mobility of PyOM potentially diverge from your observed results obtained under saturated conditions.

I appreciate the addition of fresh PyOM in the subsoil to get deeper mechanistic understanding of the dynamics of PyOM in the soil. However, besides high erosion rates and subsequent deposition, it is just hard to envisage this happening in a real setting. Not that this is a problem, you might just want to make a brief mention of it.

I am very satisfied with the methodology employed, as well as the results, tables, figures and derived conclusions. I congratulate the authors.

Specific Comments

Introduction: This is a short but well accomplished introduction. - Lines 55-57: The authors state that 'the chemical composition, physical properties and the particle size control the mobility in soils and the interactions of PyOM with the soil mineral phase

which further depend on other soil properties'. While this is true, it is important to also consider the preferential transport of fine PyOM derived from grass biomass reported elsewhere (e.g. Saiz et al, 2018). This is an important aspect considering the fine, and most likely, light nature of the PyOM used in this study, which undoubtedly will greatly affect its initial mobility after formation. (Reference: Saiz et al. 2018. Preferential Production and Transport of Grass-Derived Pyrogenic Carbon in NE-Australian Savanna Ecosystems. Frontiers in Earth Science 5, 115. doi:10.3389/feart.2017.00115)

Materials and Methods: - Line 82: The values presented in Table 1 appear to have been produced by you. If that was the case, state the methodology used to obtain them. - Line 95-on: If possible, please provide more information about the PyOM produced (i.e. O/C, H/C, etc.). This will make your work more inter-comparable with other studies. - Lines 98-103: These lines describe how PyOM was produced and, the oxidation treatment that some of those samples underwent. Please try to re-phrase these sentences as I got quite confused with the two oxidation instances that the text makes reference to. - Line 99: Table 2 shows what it seems to be a large variability between batches that have been treated in similar way. The authors may want to include some comment about it. But most importantly, if I understand well, the sandy soil gets added PyOM which is up to 10% higher in its C content compared to the PyOM that gets added to the loamy soil. Would this discrepancy not create an artifact in the behaviour of PyOM in both soils? Please critically discuss this aspect. - Line 143: Please state the nominal mesh of the glass fibre filter used.

Results: - Lines 205-207: Where can these data (statistics) be seen? - Line 231: Please check the text: '..more to than the ..' - Line 249-250: Please re-phrase this sentence. - Line 259: 'The lowest sandy subsoil layer...'. Please check this text.

Discussion: - Line 314-315: 'The first flush contributed to the highest export of PyOM from the soil columns and the mobilized amounts decreased with the percolation for all soils'. This sentence is at the beginning of a discussion section. You need to contextualize the 'flush' term a bit better. Lines 318-319: In this experiment you had

the opportunity to validate the statement about attributing the export of PyOM to mobile pyrogenic fractions directly produced during pyrolysis. Hadn't you?

Technical Comments - Line 137: Typo in 'form'. - Line 459: Typo in 'form'. - Line 482: Typo in 'desobed'.

---

## Referee Comment (RC2) · Anonymous Referee #2 · 5 Oct 2020

This paper is of high relevance, well written and provides interesting data which are certainly of interest for the readers of Biogeosciences. It is a follow-up of several previous publications, describing investigations about vertical transport in soil systems. Below, some of those publications which are not cited, but could contribute to the discussion of the present paper are mentioned. An important issue which has to be considered is the fact that there is no good distinguishing between Pyrochar (Biochar) and PyOM produced during vegetation fires. Of course both are pyrogenic organic matter, but biochar is produced under pyrolysis conditions. Such conditions may occur during peat smoldering or in subsoil fires but rarely occur during above ground fires. Although in both cases highly aromatic material is produced, there are chemical differences which may be mainly related to a more complete oxidation process during combustion in comparison to pyrolysis. This is also evidenced by the fact that combustion at 450°C is in the most cases complete and no charred remains will accumulate. However, this does not decrease the value of the present paper, since pyrolysis-derived PyOM is still PyOM and this material is in soil since it is recommended to be used as soil amendment. Therefore, I recommend to correct the definition of PyOM in Line 30 and to include Pyrochar (biochar) into this definition to make it a bit more general. As a consequence, some aspects of Pyrochar may enter the introduction. Indeed, at some places the latter is already done, although I assume that this happened unintentionally, but still a clear differentiation is needed. Below you can find some additional comments. After following those suggestions, I think the paper can be published.

31: There are many indications that the age of PyOM is by far lower than 10 000 years (Santos et al., 2012)(Hockaday et al., 2006) . Since the "real" MRT of this material is still under discussion, it should not be stated here as a proven finding. 48: Considering an atomic H/C ratio of 0.5, one cannot talk about highly condensed (Every second C is connected to a H) 51: The article by Velasco-Molina et al. (2013) is very closely related to the subject of the present paper and may be included into the discussion. 52: Change to physical and chemical, because the term phyisco-chemical is normally related to physical aspects of chemistry (i.e. thermodynamics etc.), which is definitively not the case here. 53: As mentioned above, pyrolysis is a process in which heat is applied in an oxygen-free or depleted environment. This is not the case during above ground vegetation fires. Here the vegetation is mostly combusted and the residues accumulate due to incomplete combustion (as mentioned in the introduced definition). During combustion, condensation is unlikely. In addition, the open space during a vegetation fire will decrease the probability that two volatiles can "meet" for recondensation". Only if volatilize move vertically in the soil, they may form a layer of recondensed OM. I guess the authors are referring to biochar, but this is not really clear. However, here one has to bear in mind that modern biochar production allows the removal of the syngas which prevents condensation reactions within the biochar. 59: High aromaticity is not necessarily equal to high molecular weight and it is also

not clear why high molecular weight should cause strong sorption to soil minerals. At least a reference is needed where the interested reader could get to know the included mechanisms. 98: As mentioned above, material which is pyrolyzed is not necessarily the same as material that was partially combusted. In our laboratory we have seen that material that is pyrolyzed at low temperatures ($< 500°C$) contains more alkyl C then the same residues combusted at $350°C$ (with higher temperatures complete combustion occurs). This has to be considered in the discussion. Thus, in the present work, biochar was tested rather than charcoal that is produced during a vegetation fire. 216: Most pH-meters are not exact enough to "trust" in the second post-coma digit. Thus, in the most cases it doesn't make sense to consider this digit (change to 0.2 and latter to 0.3-0.5) 293: The sentence Hilscher and Knicker… is not clear: what means "exported from the soil"? 301: The cited reference Hilscher et al. showed that PyOM from rye grass can be biochemically degraded. So why should this not be possible for the comparable material used in the present study? In the study by Velasco-Molina et al (mentioned above), the PyOM in the deeper soil horizons of a fire-prone region was highly oxidized and it was suggested that this oxidation facilitated the vertical transport. A comparable scenario may have happened here. 306: I have some problems to follow the argument. How can physical fragmentation break the bonds of an aromatic network? I think this would only work chemically. In addition, I have some problems to understand how such chemical breakdown of covalent bonds could work in soils, since such reactions need activation energy and rarely occur without catalysts or heat. What is the mechanisms behind the formation of colloids from PyOM? The authors did some Infrared on the starting material. A second analysis of the aged PyOM may deliver some more details and support the given hypothesis. 318: Again: Be careful with the term pyrolysis. This term is correct for the present paper, but does not describe the situation during natural vegetation fires described by Santos et al., or Rumpel et al., and as far as I understood, the material produced by Hilscher et al.. I also wonder about the mechanism about the production of mobile PyOM during combustion. I assume, the mobile phase would be composed of smaller molecules located within the aromatic

network of PyOM which are washed out by water. But considering the hydrophobic nature of PyOM, I wonder how water can enter the porous system of the char to do its job and how the mobile material (which should also be hydrophobic due to its aromatic nature) gets dissolved in water. I wonder if a kind of preferential flow is more likely to explain the observed vertical translocation 336: Do the authors have a proof of the statement that artificially produced PyOM is more stable than naturally produced PyOM? From a chemical viewpoint it does not make a lot of sense since the chemistry during combustion depends on the temperature and the chemistry of feedstock, irrespective of being created in the lab or in nature. Thus, PyOM of grass produced in the lab should show the same biochemical stability as material produced in nature, if it was subjected to the same temperature. However, again under natural conditions, PyOM is unlikely to have been produced at 450°C because at that temperature most of the organic matter had been volatilized. Thus, the differentiation between artificial and natural PyOM should be according to pyrolyzed and combusted charcoal. Nevertheless, I am not aware of degradation studies proofing the higher stability of "lab-made PyOM" (produced at the same temperature). If they exist, they should be referenced. 345-360: I think the results given here, are in good agreement with the study of Velasco-Molina et. al. (2013), showing that PyOM dominates the deeper horizons of a soil in a fire-prone region and that this PyOM is highly carboxylated. 365-370: A conceptual model describing this mechanism very nicely has been published in (Knicker, 2011). It may be worth to be considered in the discussion, because it already describes a considerable part of the explanations given here. There is a further very recent publication by (Miller et al., 2020), which may supportive in the discussion of the present paper. Conclusion and implementation: This part is rather a summary than a conclusion. I am missing the description of the implication the obtained results may have for our understanding of PyOM in soils. Does the mobility of PyOM affect the aquifer? Is there an environmental threat? What may happen based on the findings of the present paper?

Hockaday, W.C., Grannas, A.M., Kim, S., Hatcher, P.G., 2006. Direct molecular evidence for the degradation and mobility of black carbon in soils from ultrahighresolution mass spectral analysis of dissolved organic matter from a fire-impacted forest soil. Org. Geochem. 37, 501–510. Knicker, H., 2011. Pyrogenic organic matter in soil: Its origin and occurrence, its chemistry and survival in soil environments. Quat. Int. 243. https://doi.org/10.1016/j.quaint.2011.02.037 Miller, A.Z., De la Rosa, J.M., Jiménez-Morillo, N.T., Pereira, M.F.C., Gonzalez-Perez, J.A., Knicker, H., Saiz-Jimenez, C., 2020. Impact of wildfires on subsurface volcanic environments: New insights into speleothem chemistry. Sci. Total Environ. 698, 134321. https://doi.org/https://doi.org/10.1016/j.scitotenv.2019.134321 Santos, F., Torn, M.S., Bird, J.A., 2012. Biological degradation of pyrogenic organic matter in temperate forest soils. Soil Biol. Biochem. 51, 115–124. https://doi.org/http://dx.doi.org/10.1016/j.soilbio.2012.04.005 Velasco-Molina, M., Knicker, H., Macías, F., 2013. The potential of humic material in sombric-like horizons of two brazilian soil profiles as an efficient carbon sink within the global C cycle, Functions of Natural Organic Matter in Changing Environment. https://doi.org/10.1007/978-94-007-5634-2_78

---

## Author Comment (AC1) · 15 Oct 2020

You find our final author responses written in blue.

General Comments
This manuscript provides information on the vertical mobility of pyrogenic organic matter in soils of contrasting nature. The study uses a column experiment where isotopically labelled char gets added to soils collected at different depths. The movement of both added PyOM and native soil is then traced through the soil column. The study is very relevant, well designed and impeccably executed. The MS is well written and provides a pleasant reading. It has been a few years since I did not enjoy reading a first submission as much as I have with this one. I have little doubt that this work will prove quite useful for researchers studying PyOM dynamics and ecosystem carbon cycling.

We are grateful and appreciate the positive feedbacks on our submitted manuscript.

There are just a few aspects where I would appreciate that the authors provide further critical discussion to make this contribution an even more useful one. These are as follows:
The PyOM used in this study derives from ryegrass. I can understand why such fastgrowing precursor biomass was used to produce PyOM in this labelled experiment. However, it is unavoidable to think that the resultant pyrogenic material will be of a highly contrasting physico-chemical nature compared to those derived from woody vegetation. Therefore, these distinct characteristics may greatly affect the mobility of the various PyOM produced. As such, I recommend the authors to include a paragraph in the discussion showcasing the potential limitations and applicability of the results obtained in this study. The results obtained here may be directly applicable in agronomic studies using grass-derived biochar. However, the mobility of PyOM in charcoal generated during wildfires affecting woody vegetation might be different from that observed in grass-derived PyOM.

We agree with the reviewer that this is an important aspect, which needs to be included to indicate the limitations of this experimental set-up. The ryegrass in our experiment is mainly used because it is much easier to produce with high label than wood. A lower label however, would constrain the tracing of the mobilized PyOM because of its quite low mobile fraction.
In the present manuscript we already mentioned this limitation in line 302-303 by stating: "It needs to be noticed that we only included one type of PyOM (ryegrass derived and produced at 450°C) which constrains general assumptions". However, we will follow the suggestion and highlight this important limitation again in the discussion and include it further in the conclusion. In general, grassland systems represent a major source of fire derived organic matter globally, so the use of grass material is still legitimate.

The existence of fluctuating levels of moisture in the soil is just natural. Please briefly include an statement about how soil drying and wetting events may cause the mobility of PyOM potentially diverge from your observed results obtained under saturated conditions.

We agree that unsaturated flow conditions are highly relevant for the mobility of PyOM under field conditions. Based on the findings from our experimental set-up, we cannot provide any statement how the mobility would be influenced under unsaturated conditions.
We found a quick interaction of the PyOM with the mineral phase even under saturated conditions in our experiment. Unsaturated conditions and unsaturated water flow could potentially increase the contact time of PyOM and the mineral phase and thus influencing the interaction with each other but it could also increase the preferential flow and thus increase the downward movement with less interaction with the mineral phase. This would also highly depend on the texture and aggregation of the soil. Therefore, this requires more research and we stated in line 392-393 of the current manuscript: "This quick mineral interaction will control the long-term stability of PyOM in soils and requires more research under unsaturated and field conditions."
We also discussed that our experimental approach has limitations that prevent a direct transfer to field conditions due to the continuous saturated flow conditions in our experiment and in line 408-409 of the current manuscript, we state: "In addition, we did not include unsaturated flow conditions which would resemble the mobility and retention under field conditions".

I appreciate the addition of fresh PyOM in the subsoil to get deeper mechanistic understanding of the dynamics of PyOM in the soil. However, besides high erosion rates and subsequent deposition, it is just hard to envisage this happening in a real setting. Not that this is a problem, you might just want to make a brief mention of it.

We agree with the reviewer that the input of fresh PyOM in subsoils maybe not be common under field conditions. We stated this also in the beginning of our introduction in line 34-36 of the current manuscript: "Mass transport of PyOM mainly occurs during the first rain event after a fire, resulting in a translocation and re-deposition within the landscape and eventually in a PyOM burial at depositional sites (Abney et al., 2019; Cotrufo et al., 2016; Rumpel et al., 2015)." This burial may happen before the PyOM is significantly oxidized on its surfaces.
We further address this, when we discuss the dynamic of mobilized PyOM in section 4.2. by stating in line 323-323: "This may further cause an underestimation of the vertical PyOM transport from depositional landscape positions after redistribution following an initial lateral mass transport (Rumpel et al., 2015)." We will highlight again that depositional sites could represents location were PyOM may have entered subsoil regions and add a sentence in the discussion.

I am very satisfied with the methodology employed, as well as the results, tables, figures and derived conclusions. I congratulate the authors.

We highly appreciate the reviewers positive feedbacks.

Specific Comments
Introduction: This is a short but well accomplished introduction.

 - Lines 55-57: The authors state that 'the chemical composition, physical properties and the particle size control the mobility in soils and the interactions of PyOM with the soil mineral phase which further depend on other soil properties'. While this is true, it is important to also consider the preferential transport of fine PyOM derived from grass biomass reported elsewhere (e.g. Saiz et al, 2018). This is an important aspect considering the fine, and most likely, light nature of the PyOM used in this study, which undoubtedly will greatly affect its initial mobility after formation. (Reference: Saiz et al. 2018. Preferential Production and Transport of Grass-Derived Pyrogenic Carbon in NE-Australian Savanna Ecosystems. Frontiers in Earth Science 5, 115. doi:10.3389/feart.2017.00115)

We appreciate that the reviewer provided further literature to highlight this important aspect. We also agree that this needs to be clearly mentioned and we will improve the mentioned section. Saiz's reference will be added to the manuscript.

Materials and Methods:
- Line 82: The values presented in Table 1 appear to have been produced by you. If that was the case, state the methodology used to obtain them.

All these values were measured by us. The methods, which were used for the bulk soil measurements, are given in section 2.4. To clarify this, we will refer to this section in the caption of table 1 and change to: "Table 1: Soil texture, total organic carbon (TOC), $\delta 13C$, pH, electrical conductivity (EC), oxalate extractable Fe(o) and Al(o) and density fractions (free particulate organic matter, fPOM and mineral associated organic matter, MAOM) for the topsoil (0-10 cm depth) and subsoil (40-60 cm) of the loamy and sandy soil (± 1 SE). See section 2.4 for the used methods."

- Line 95-on: If possible, please provide more information about the PyOM produced (i.e. O/C, H/C, etc.). This will make your work more inter-comparable with other studies.

We are currently accessing the O/C and H/C ratios by additional CNHS O analysis and we will include these parameters in Table 2.

- Lines 98-103: These lines describe how PyOM was produced and, the oxidation treatment that some of those samples underwent. Please try to re-phrase these sentences as I got quite confused with the two oxidation instances that the text makes reference to.

We agree that the current description may not be clear and we will rephrase to the following: "Artificially altered PyOM was produced by chemical and heat accelerated oxidation presented by Cross and Sohi (2013); in brief, 1 g of C was oxidized with 0.1 mol of $H_2O_2$ at 80°C for two days. The samples were gently shaken five to seven times a day to ensure a homogeneous reaction."

- Line 99: Table 2 shows what it seems to be a large variability between batches that have been treated in similar way. The authors may want to include some comment about it. But most importantly, if I understand well, the sandy soil gets added PyOM which is up to 10% higher in its C content compared to the PyOM that gets added to the loamy soil. Would this discrepancy not create an artifact in the behaviour of PyOM in both soils? Please critically discuss this aspect.

We noticed that a variability between the produced ryegrass batches exist. Such variation is hard to explain since the growing conditions were controlled and similar for all three batches. However, it is possible that the uptake of, for example, silica differs slightly between the batches which would result in variations of the C content. However, all the recovery calculations (total C and 13C) are based on the actual values presented in Table 2 and thus the variability is taken into account.

We agree that this needs to be mentioned in the manuscript and we will include in section 2.2 the following: "It needs to be noticed that the three batches of ryegrass PyOM varied in C content and $\delta^{13}$C. To include this variation, all further calculations are based on the individual values (Table 2)."

 - Line 143: Please state the nominal mesh of the glass fibre filter used.

Will be changed to: "…filtered using a glass fiber filter (<0.7 μm)."

Results:
- Lines 205-207: Where can these data (statistics) be seen?

The data can be seen in figure 2. We choose the same scaling between the topsoils and subsoils as well as between the sandy and loamy soil to allow a direct comparison of the percolated PyOM. We will refer to Fig 2 again.

- Line 231: Please check the text: '..more to than the ..'

We will correct this sentence to: "Thus, PyOM addition significantly increased the nSOC leaching for the sandy topsoil the control without PyOM addition, from which 1.7 ± 0.1 % of the initial nSOC were leached (p<0.01)"

- Line 249-250: Please re-phrase this sentence.

We will rephrase the sentence to: "After the percolation, 89-96 % of the fresh and oxidized PyOM-C remained at its initial location in the PyOM layer in both soils."

- Line 259: 'The lowest sandy subsoil layer. . .'. Please check this text.

We will specify and rephrase to: "In 4.6-7.0 cm depth below the PyOM layer, the recoveries of oxidized (p<0.05) and fresh PyOM-C (p<0.01) were significantly higher in the sandy subsoils compared to the topsoil.

Discussion:
- Line 314-315: 'The first flush contributed to the highest export of PyOM from the soil columns and the mobilized amounts decreased with the percolation for all soils'. This sentence is at the beginning of a discussion section. You need to contextualize the 'flush' term a bit better.

We specified the first flush as the first 1,000 l m$^{-2}$ in the result section. We will specify it here again by adding: "The first flush, leached form the column with 1,000 l m$^{-2}$, …"

-Lines 318-319: In this experiment you had the opportunity to validate the statement about attributing the export of PyOM to mobile pyrogenic fractions directly produced during pyrolysis. Hadn't you?

The production of large quantities of highly labelled PyOM is very challenging. Therefore, we do not have enough material left to determine water extractable fractions with a sufficient certainty.

Technical Comments
- Line 137: Typo in 'form'.

We will correct and carefully check the manuscript again for similar typos

- Line 459: Typo in 'form'.

Same as above

- Line 482: Typo in 'desobed'.

Same as above

---

## Author Comment (AC2) · 15 Oct 2020

You find our final authors responses written in blue

General comments:
This paper is of high relevance, well written and provides interesting data which are certainly of interest for the readers of Biogeosciences. It is a follow-up of several previous publications, describing investigations about vertical transport in soil systems. Below, some of those publications which are not cited, but could contribute to the discussion of the present paper are mentioned. An important issue which has to be considered is the fact that there is no good distinguishing between Pyrochar (Biochar) and PyOM produced during vegetation fires. Of course both are pyrogenic organic matter, but biochar is produced under pyrolysis conditions. Such conditions may occur during peat smoldering or in subsoil fires but rarely occur during above ground fires. Although in both cases highly aromatic material is produced, there are chemical differences which may be mainly related to a more complete oxidation process during combustion in comparison to pyrolysis. This is also evidenced by the fact that combustion at 450°C is in the most cases complete and no charred remains will accumulate. However, this does not decrease the value of the present paper, since pyrolysis-derived PyOM is still PyOM and this material is in soil since it is recommended to be used as soil amendment. Therefore, I recommend to correct the definition of PyOM in Line 30 and to include Pyrochar (biochar) into this definition to make it a bit more general. As a consequence, some aspects of Pyrochar may enter the introduction. Indeed, at some places the latter is already done, although I assume that this happened unintentionally, but still a clear differentiation is needed. Below you can find some additional comments. After following those suggestions, I think the paper can be published.

We thank the reviewer for the general positive feedbacks and for the useful suggestions for our manuscript. We agree that we need to include a better definition and differentiation between artificially pyrolyzed and natural PyOM which is a product from wildfires. Therefore, we will change the first sentence of the introduction to: "Pyrogenic organic matter (PyOM) is a product of artificial (biochar) or wildfire induced incomplete combustion.". We will also follow the suggestions below to state clearly that we used artificially pyrolyzed PyOM in our experiment and highlight the associated limitations.

Specific comments:

31: There are many indications that the age of PyOM is by far lower than 10 000 years (Santos et al., 2012)(Hockaday et al., 2006) . Since the "real" MRT of this material is still under discussion, it should not be stated here as a proven finding.

The first section of the introduction of the current manuscript provides an overview of the PyOM in the Earth system before it narrows down to the soil system. Therefore, the given residence time of >10,000, which is reported in the cited literature (e.g. Coppola et al 2018; Masiello and Druffel 1998), represents a residence time of PyOM in the Earth system. If we are not mistaken, the reviewer refers, with the suggested study from Santos et al. (2012), to the residence time in soils. We start to discuss the current knowledge of PyOM residence times in soils from the paragraph line 46-51 and following. Here we state that the MRT of PyOM is higher than the MRT of non-pyrogenic soil organic matter (line 47). However, we agree that this is still an open discussion and PyOM is also reported with lower MRT in soils. Thus, we will change line 47 as followed and include the suggested literature: "Pyrogenic organic matter is found with residence times in soils of several centuries to millennia, which is much more than the average age of SOC and is mainly attributed to its condense and aromatic composition and thus increased stability against degradation (Kuzyakov et al., 2014; Santos et al., 2012; Schmidt et al., 2011; Singh et al., 2012)."
The suggested literature, Hockaday et al. (2006), is already cited and included in our discussion (line 352-355). Since this publication is not reporting MRT, we think it is better not to cite it there, but, we agree that this study should be included in the introduction. We will add in line 66: "In addition, it is reported that PyOM found in fire effected watersheds underwent aging processes in soil prior to its export from the soils to the riverine system (Hockaday et al., 2006)."

48: Considering an atomic H/C ratio of 0.5, one cannot talk about highly condensed (Every second C is connected to a H)

We are sorry for the misunderstanding. We did not mention in our text a H/C ratio of 0.5 and we are not addressing any aspect directly related to specific degree of aromaticity or condensed structure here. We will remove the "highly" in line 47 to avoid confusion. We already included this change in our response to the previous comment. Furthermore, we will include the actual H/C and O/C ratios in Table 2 as it was recommended by reviewer 1. We are currently measure the H and O content but first results indicate that the H/C ratios are rather ranging between 0.1-0.2, indicating a higher degree of condensation than mentioned by the reviewer.

51: The article by Velasco-Molina et al. (2013) is very closely related to the subject of the present paper and may be included into the discussion.

We agree that the study from Velasco-Molina et al. (2013) is supporting the assumption that vertical transport in soils is determine the long-term fate of PyOM in soils. However, we cite a very recent review from Hobley (2019) which already includes the mentioned study. In order, to be consistent, we will remove the citation of Foereid et al. (2011) and Leifeld et al. (2007) because these are also discussed in the review from Hobley (2019).

52: Change to physical and chemical, because the term phyisco-chemical is normally related to physical aspects of chemistry (i.e. thermodynamics etc.), which is definitively not the case here.

Will be changed as recommended by the reviewer

53: As mentioned above, pyrolysis is a process in which heat is applied in an oxygen-free or depleted environment. This is not the case during above ground vegetation fires. Here the vegetation is mostly combusted and the residues accumulate due to incomplete combustion (as mentioned in the introduced definition). During combustion, condensation is unlikely. In addition, the open space during a vegetation fire will decrease the probability that two volatiles can "meet" for recondensation". Only if volatilize move vertically in the soil, they may form a layer of recondensed OM. I guess the authors are referring to biochar, but this is not really clear. However, here one has to bear in mind that modern biochar production allows the removal of the syngas which prevents condensation reactions within the biochar.

We agree that condensation can be interpreted in different ways and we do not provide a clear definition here. To avoid confusion, we will change line 53 to: "Pyrolysis affects the chemical and physical properties of the feedstock organic matter which result in a high porosity and large surface areas depending on the fuel biomass, duration and production temperature (Hammes and Abiven, 2013; Lehmann et al., 2015; Preston and Schmidt, 2006).". Further, we will include an additional reference which we missed in the current manuscript but supports the given statement: Lehmann, J., Abiven, S., Kleber, M., Pan, G., Singh, B. P., Sohi, S. P., & Zimmerman, A. R. (2015). Persistence of biochar in soil. In J. Lehmann & S. D. Joseph (Eds.), Biochar for Environmental Management (Issue January, pp. 235–282). Routledge. https://doi.org/10.4324/9780203762264-17

59: High aromaticity is not necessarily equal to high molecular weight and it is also not clear why high molecular weight should cause strong sorption to soil minerals. At least a reference is needed where the interested reader could get to know the included mechanisms.

We are sorry for the misunderstanding. We did not want to say that high aromaticity is equal to a high molecular weight here. We state that PyOM is rich in aromatic compounds and thus will most probably have a higher sorption affinity to the mineral surface. This is discussed in the discussion section (line 450). We agree that this statement requires further references and we will include here: Kaiser, K., & Guggenberger, G. (2000). The role of DOM sorption to mineral surfaces in the preservation of organic matter in soils. Organic Geochemistry, 31(7–8), 711–725. The authors identified that the sorption of organic matter is depending on its chemical structure and compounds containing aromatic structures (such as lignin) have a higher sorption affinity than less aromatic compounds.

98: As mentioned above, material which is pyrolyzed is not necessarily the same as material that was partially combusted. In our laboratory we have seen that material that is pyrolyzed at low temperatures (< 500°C) contains more alkyl C then the same residues combusted at 350°C (with higher temperatures complete combustion occurs). This has to be considered in the discussion. Thus, in the present work, biochar was tested rather than charcoal that is produced during a vegetation fire.

We agree that partially combusted and artificially pyrolyzed PyOM differ in several properties. Actually, we already included this in our current discussion line 336:" Artificially produced PyOM is mostly more stable than naturally produced PyOM, which challenges the use of one type as a proxy for the other (Santín et al., 2017).". We agree that this needs to be addressed again in the mentioned section. Therefore, we will change this to: "The ryegrass was oven dried at 40°C and pyrolyzed at 450°C for 4 h under $N_2$ atmosphere (Hammes et al., 2006). Three

independent growing batches of the initial ryegrass were pyrolyzed separately which were used in our experiment as a proxy for PyOM (Table 2)."

216: Most pH-meters are not exact enough to "trust" in the second post-coma digit. Thus, in the most cases it doesn't make sense to consider this digit (change to 0.2 and latter to 0.3-0.5)

Will be changed as recommended

293: The sentence Hilscher and Knicker. . . is not clear: what means "exported from the soil"?

Hilscher and Knicker (2011) reported that 0.4% of initially applied PyOM were found in the outflow of soil columns (8cm length) in a one-year incubation experiment. We will clarify this here and change to: "Hilscher and Knicker (2011) reported that 2.3 % of added PyOM migrated to 5 cm depth and 0.4 % were leached from soil columns (8 cm length) and found in the column outflow in a one-year incubation experiment."

301: The cited reference Hilscher et al. showed that PyOM from rye grass can be biochemically degraded. So why should this not be possible for the comparable material used in the present study? In the study by Velasco-Molina et al (mentioned above), the PyOM in the deeper soil horizons of a fire-prone region was highly oxidized and it was suggested that this oxidation facilitated the vertical transport. A comparable scenario may have happened here.

We agree that biochemical degradation and the associated oxidation of PyOM is an important factor controlling the vertical PyOM mobility in soils. This was also clearly shown with our experiment and the higher mobility and reactivity of the oxidized PyOM. In fact, we designed the experiment according to the existing information in the literature which reported higher mobility of aged and oxidized PyOM. In order to include aged PyOM, we used the accelerated aging as described by Cross and Sohi (2013). Due to the comparable short duration of our experiment (five days of percolation - line 119-121), we can assume that an additional biochemical degradation of the PyOM is negligible.

306: I have some problems to follow the argument. How can physical fragmentation break the bonds of an aromatic network? I think this would only work chemically. In addition, I have some problems to understand how such chemical breakdown of covalent bonds could work in soils, since such reactions need activation energy and rarely occur without catalysts or heat. What is the mechanisms behind the formation of colloids from PyOM? The authors did some Infrared on the starting material. A second analysis of the aged PyOM may deliver some more details and support the given hypothesis.

The physical fractionation discussed here and also discussed in the cited references (Hobley, 2019; Pignatello et al., 2015) takes place on a larger scale than mentioned by the reviewer. In the above references, it is reported that centimetric to millimetric PyOM particles will breakdown into smaller particles (sub millimetric) due to physical breakdown with time. This is a process happening rather on the scale of cm to µm than at the molecular scale.
The formation of colloids and PyOM particles aggregation is controlled by the surface interaction of the PyOM. According the cited literature in line 308 (Castan et al., 2019; Sigmund et al., 2018), the aggregation is controlled by van der Waals attraction and electrostatic repulsion.
The mid-infrared spectra of the fresh and oxidized PyOM showed an increase in functional groups with oxidation (Section 2.2 and Fig. 1). As mentioned above, further degradation should be negligible due to the short duration of the experiment, so we did not measure the mid-infrared spectrum of the PyOM after the experiment.

318: Again: Be careful with the term pyrolysis. This term is correct for the present paper, but does not describe the situation during natural vegetation fires described by Santos et al., or Rumpel et al., and as far as I understood, the material produced by Hilscher et al.. I also wonder about the mechanism about the production of mobile PyOM during combustion. I assume, the mobile phase would be composed of smaller molecules located within the aromatic network of PyOM which are washed out by water. But considering the hydrophobic nature of PyOM, I wonder how water can enter the porous system of the char to do its job and how the mobile material (which should also be hydrophobic due to its aromatic nature) gets dissolved in water. I wonder if a kind of preferential flow is more likely to explain the observed vertical translocation

We agree that the term pyrolysis is correct for the PyOM used in our experiment. Our experiment did not allow to specifically investigate the mechanism of the PyOM mobilization by the percolating water because we determined the mobile fraction as a sum of dissolved and particulate PyOM. Our breakthrough curve analysis, however, indicate the absence of preferential flow (See also supplement material) and that the application of PyOM did not change the hydraulic properties of the soil column compared to the controls.
Santos et al. (2017) concluded with their field observations that also natural PyOM may contain easily mobilized fractions which are mobilized with the first flush after the fire. However, the quantification of this initial pulse of PyOM flux was up to now mostly missing in the literature. We were able to capture this first pulse of mobilized PyOM in our experiment, which contributed to up to 84% of the total mobilized PyOM.

336: Do the authors have a proof of the statement that artificially produced PyOM is more stable than naturally produced PyOM? From a chemical viewpoint it does not make a lot of sense since the chemistry during combustion depends on the temperature and the chemistry of feedstock, irrespective of being created in the lab or in nature. Thus, PyOM of grass produced in the lab should show the same biochemical stability as material produced in nature, if it was subjected to the same temperature. However, again under natural conditions, PyOM is unlikely to have been produced at 450_C because at that temperature most of the organic matter had been volatilized. Thus, the differentiation between artificial and natural PyOM should be according to pyrolyzed and combusted charcoal. Nevertheless, I am not aware of degradation studies proofing the higher stability of "lab-made PyOM" (produced at the same temperature). If they exist, they should be referenced.

The cited study by Santín et al (2017) provides an extensive and direct chemical comparison of natural and artificially pyrolyzed PyOM produced at similar temperatures and similar feedstock (e.g. degree of aromaticity, H:C, O:C, recalcitrance index). The authors identified that slow-pyrolysis compared to the fast combustion during wildfires resulted in a higher stability of the artificial PyOM. Therefore, the authors concluded that the use of PyOM from pyrolysis as a proxy for natural PyOM may be limited.
In order to be more specific, we will change the mentioned line 336-337 to: "Artificially produced PyOM is reported to have a higher stability than PyOM naturally produced during wildfires, which challenges the use of one type as a proxy for the other (Santín et al., 2017)."

345-360: I think the results given here, are in good agreement with the study of Velasco-Molina et. al. (2013), showing that PyOM dominates the deeper horizons of a soil in a fireprone region and that this PyOM is highly carboxylated.

We thank the reviewer for providing the additional literature which we will integrate to support our findings with field observations described by Velasco-Molina et al. (2013)

365-370: A conceptual model describing this mechanism very nicely has been published in (Knicker, 2011). It may be worth to be considered in the discussion, because it already describes a considerable part of the explanations given here. There is a further very recent publication by (Miller et al., 2020), which may supportive in the discussion of the present paper.

We thank the reviewer for further literature to strengthen our discussion and to include the existing conceptual model described by Knicker (2011).

Conclusion and implementation: This part is rather a summary than a conclusion. I am missing the description of the implication the obtained results may have for our understanding of PyOM in soils. Does the mobility of PyOM affect the aquifer? Is there an environmental threat? What may happen based on the findings of the present paper?

We think that a discussion of direct effects on aquifers and environmental threats would be too speculative with our small scale experiment. This would require more research under field conditions and on larger scale. Therefore, we will emphasis this by adding the following: "Further research is needed to understand the fate of PyOM under unsaturated and field conditions and larger scales such as pedon and catena. We identified that the vertical PyOM mobility is highly depending on soil properties and the degree of PyOM oxidation (age) which increases not only its mobility, but also reactivity in soils and influences its effect on nSOC. This will influence its dynamics in the vadose zone and between the terrestrial and aquatic systems."

Suggested literature:
Hockaday, W.C., Grannas, A.M., Kim, S., Hatcher, P.G., 2006. Direct molecular evidence for the degradation and mobility of black carbon in soils from ultrahigh- resolution mass spectral analysis of dissolved organic matter from a fire-impacted forest soil. Org. Geochem. 37, 501–510.

Knicker, H., 2011. Pyrogenic organic matter in soil: Its origin and occurrence, its chemistry and survival in soil environments. Quat. Int. 243. https://doi.org/10.1016/j.quaint.2011.02.037

Miller, A.Z., De la Rosa, J.M., Jiménez-Morillo, N.T., Pereira, M.F.C., Gonzalez-Perez, J.A., Knicker, H., Saiz-Jimenez, C., 2020. Impact of wildfires on subsurface volcanic environments: New insights into speleothem chemistry. Sci. Total Environ.698, 134321. https://doi.org/https://doi.org/10.1016/j.scitotenv.2019.134321

Santos, F., Torn, M.S., Bird, J.A., 2012. Biological degradation of pyrogenic organic matter in temperate forest soils. Soil Biol. Biochem. 51, 115–124. https://doi.org/http://dx.doi.org/10.1016/j.soilbio.2012.04.005

Velasco-Molina, M., Knicker, H., Macías, F., 2013. The potential of humic material in sombric-like horizons of two brazilian soil profiles as an efficient carbon sink within the global C cycle, Functions of Natural Organic Matter in Changing Environment. https://doi.org/10.1007/978-94-007-5634-2_78

Literature which will be added

Kaiser, K., & Guggenberger, G. (2000). The role of DOM sorption to mineral surfaces in the preservation of organic matter in soils. *Organic Geochemistry*, *31*(7–8), 711–725. https://doi.org/10.1016/S0146-6380(00)00046-2

---

## Author Response (AR2)

Authors responses

**Editor comments on "Vertical mobility of pyrogenic organic matter in soils: A column experiment" by Marcus Schiedung et al.**

You find our authors responses written in blue

Many thanks for revising the manuscript, which I think has improved following the input of both referees. I have two remaining significant points that I would like you to address before I make a final decision regarding acceptance.

Line 33: I share referee 2's concern regarding the very general statement about PyC residence times of being "on the order of 10,000 years". Whilst this is within the range of measured residence times, the actual mean turnover of the PyC pool in terrestrial ecosystems is lower. Coppola et al refer to ages of "up to 17,000 years", while Masiello & Druffel present data from specific deep sea sediments, and even there residence times are above 10,000 years only at one site. Your statement

15 here needs qualifying as the long residence time suggests a mean value that is not correct for the majority of conditions. You later illustrate this in lines 49/50, so I'm not sure why this very general (and in its brevity misleading) statement is provided here.

We agree that the mentioned section can still be misleading after our last revision. We changed it to the following and hope to avoid any further confusion: "It is one of the oldest global organic carbon (C) pools with residence times of several

20 millennia (Coppola et al., 2018; Masiello and Druffel, 1998) and an annual global production from wildfires of 196-349 Tg C (Jones et al., 2019)"

A further issue that needs to be addressed is the very high rate of percolation in your experiment. The initial flush is equivalent of 1000 l per m2, or 1000 mm of precipitation, significantly more than annual rainfall at both your sites. The total

25 amount of flow through your columns (18,000 l m-2) is equivalent of 25 years of precipitation, which you applied over only 5 days. Please address this issue in the discussion. Even if the lab set-up can't simulate natural field conditions, I think that the choice of a magnified flow regime on this scale has to be justified, and the impact of PyC transport under more natural conditions discussed.

We agree that the flow rates applied in our experiment need to be included again in the discussion. Based on the suggested of

30 reviewer 1 and 2, we improved the initial manuscript by highlighting the saturated flow conditions in our experiment as the major constraints to directly relate our observation with field conditions. To avoid any direct relation to filed conditions, we also decided not to include any further interpretation of years of precipitation in the initial manuscript. But we see that this would be needed to be discussed at one point. Therefore, we included in Line 321: "We percolated the soil columns in total with 18,000 l m-2, which would be equal to a continuous precipitation of 23-29 years given the mean annual precipitation of

35    the two sites (sandy soil=620 mm and loamy soil=780 mm). Therefore, the percolation applied here over 5 days was conducted at relatively high rates. This experimental duration avoided any additional decomposition within the columns and allowed to maintain a continuously saturated soil column system. However, given the experimental set-up, we were not able to estimate any PyOM transportation rate under field conditions under which the transport is influenced by seasonal precipitation variations and unsaturated conditions."

40

I also have a short list of relatively minor edits, and hope that you are happy to include these:

Line 48: Please delete "to" before "one".

Changes as suggested

45

Line 50: "condensed"

Change as suggested

Line 54: put comma after brackets and delete "and"

50    Change as suggested

Line 60: comma after "phase"

Changes as suggested

55    Line 71: "affected" rather than "effected".

Changed as suggested

Line 317: I think you refer to "general conclusions" here, rather than "general assumptions". Please revise.

Changed as suggested

60

Table captions (both T1 and T2): please delete "the used".

Changed as suggested

Table 2: Please round d13C values to whole numbers (i.e. 4 significant figures); error terms with one decimal place are ok.

65    Changed as suggested but we also changed to standard error to whole numbers

[revised manuscript text omitted]